# The Polycomb group protein Ring1 regulates dorsoventral patterning of the mouse telencephalon

Hikaru Eto[1,6], Yusuke Kishi [1,6✉], Nayuta Yakushiji-Kaminatsui [2], Hiroki Sugishita[2], Shun Utsunomiya[1,3,5], Haruhiko Koseki[2] & Yukiko Gotoh [1,4✉]

Dorsal-ventral patterning of the mammalian telencephalon is fundamental to the formation of distinct functional regions including the neocortex and ganglionic eminence. While Bone morphogenetic protein (BMP), Wnt, and Sonic hedgehog (Shh) signaling are known to determine regional identity along the dorsoventral axis, how the region-specific expression of these morphogens is established remains unclear. Here we show that the Polycomb group (PcG) protein Ring1 contributes to the ventralization of the mouse telencephalon. Deletion of *Ring1b* or both *Ring1a* and *Ring1b* in neuroepithelial cells induces ectopic expression of dorsal genes, including those for BMP and Wnt ligands, as well as attenuated expression of the gene for Shh, a key morphogen for ventralization, in the ventral telencephalon. We observe PcG protein–mediated trimethylation of histone 3 at lysine-27 and binding of Ring1B at BMP and Wnt ligand genes specifically in the ventral region. Furthermore, forced activation of BMP or Wnt signaling represses Shh expression. Our results thus indicate that PcG proteins suppress BMP and Wnt signaling in a region-specific manner and thereby allow proper Shh expression and development of the ventral telencephalon.

[1] Graduate School of Pharmaceutical Sciences, The University of Tokyo, 7-3-1 Hongo, Bunkyo-ku, Tokyo 113-0033, Japan. [2] Laboratory for Developmental Genetics, RIKEN Center for Integrative Medical Sciences (RIKEN-IMS), 1-7-22, Suehiro-cho, Tsurumi-ku, Yokohama 230-0045, Japan. [3] Graduate School of Engineering, The University of Tokyo, 7-3-1 Hongo, Bunkyoku, Tokyo 113-0033, Japan. [4] International Research Center for Neurointelligence (WPI-IRCN), The University of Tokyo, 7-3-1 Hongo, Bunkyo-ku, Tokyo 113-0033, Japan. [5]Present address: Neuroscience 2, Drug Discovery & Disease Research Laboratory, Shionogi & Co., Ltd.; Business-Academia Collaborative Laboratory (Shionogi), Graduate School of Pharmaceutical Sciences, The University of Tokyo, 7-3-1 Hongo, Bunkyo-ku, Tokyo 113-0033, Japan. [6]These authors contributed equally: Hikaru Eto, Yusuke Kishi. ✉email: ykisi@mol.f.u-tokyo.ac.jp; ygotoh@mol.f.u-tokyo.ac.jp

In vertebrate embryos, the telencephalon is formed at the most anterior portion of the developing central nervous system (CNS). The cerebral cortex (CTX) and ganglionic eminence (GE) are derived from the dorsal telencephalon (the pallium) and the ventral telencephalon (the subpallium), respectively[1,2]. The dorsal midline of the telencephalon contains the hem, which induces dorsal patterning including the formation of the hippocampus and its related regions. Regulation of dorsal-ventral (D-V) patterning is thus fundamental to the development of the telencephalon.

The regional identity of telencephalic neural stem-progenitor cells (NPCs) along the D-V axis is determined by region-specific transcription factors (TFs) such as Pax6 and Emx1/2 in the CTX, Gsx2 in the lateral and medial GE (LGE and MGE, respectively), and Nkx2.1 in the MGE[3–6]. Mutual gene repression by Pax6 and Gsx2 contributes to the establishment of the D-V boundary[5]. Neurog1 and Neurog2, proneural genes in the CTX, and Ascl1, a proneural (and oligodendrogenic) gene in the GE, are expressed according to the regional identity of NPCs in mouse embryos[7–9].

Telencephalic regionalization along the D-V axis begins before the closure of the neural tube, which occurs around embryonic day (E) 9.0 in mice, and is established before the onset of neurogenesis at ~E10. The initial stages of D-V patterning are controlled by secreted morphogenetic signals (morphogens) that spread over various distances. The combination of the activities of different morphogens gives rise to distinct expression patterns of region-specific TFs in the telencephalon[10–13], as is also the case in the vertebrate spinal cord[14] and in invertebrate embryos[15]. Bone morphogenetic proteins (BMPs), Wnt ligands, Sonic hedgehog (Shh), and fibroblast growth factor 8 (FGF8) are among the morphogens involved in D-V patterning in the mammalian telencephalon.

The dorsal midline regulates the dorsal patterning of the telencephalon[16]. BMPs (BMP4, −5, −6, and −7) are secreted from the dorsal midline and paramedial neuroectoderm in the prospective forebrain[17] and play pivotal roles in such patterning through induction of target genes including Msx1, Lmx1a, and Wnt3a[17–21]. Knockout of BMP receptors thus results in loss of the dorsal-most structures of the telencephalon including the cortical hem and choroid plexus[20]. Wnt ligands are also expressed in the dorsal region (Wnt1, −3, −3a, and −7b in the dorsal telencephalic roof plate at E9.5; Wnt2b, −3a, −5a, −7b, and 8b in the cortical hem and Wnt7a and −7b in the CTX at later stages) and contribute to aspects of dorsal patterning such as the formation of the cortical hem and the CTX through induction of various TFs including Lef1 as well as Emx1/2, Pax6, and Gli3, respectively[11,22,23]. In addition, Wnt signaling increases the activity of Lhx2, a selector gene for the CTX[24]. Suppression of Wnt signaling increases expression of ventral region-specific genes throughout the dorsal pallium, indicating the importance of such signaling in dorsal patterning[22].

Shh, on the other hand, plays a major role in ventral patterning of the telencephalon[25]. Shh is secreted initially from the anterior mesendoderm or the prechordal plate[26], then from the ventral hypothalamus, and finally from the rostroventral telencephalon including the preoptic area and MGE[27–29]. Shh signaling induces the expression of ventral TFs such as Foxa2, Nkx2.1, and Gsx2 and suppresses the repressor activity of Gli3, which is crucial for development of the dorsal telencephalon[30–32].

Expression of BMP ligands and activation of BMP signaling are confined to the dorsal midline, with this confinement being critical for the development of the ventral telencephalon, given that ectopic BMP signaling can suppress the expression of ventral morphogenetic factors such as Shh and FGF8 as well as that of the ventral TF Nkx2.1 in the chick forebrain[33]. It is also important that Wnt signaling be confined to the dorsal pallium, given that ectopic activation of such signaling suppresses ventral specification in the developing mouse telencephalon[22]. The mechanism responsible for the regional confinement of these dorsal morphogens in the early stage of telencephalic development has remained largely unclear, however.

Polycomb group (PcG) proteins are repressive epigenetic factors that form two complexes, PRC1 and PRC2. These complexes catalyze the ubiquitylation of histone 2A at lysine-119 (H2AK119ub) and the trimethylation of histone 3 at lysine-27 (H3K27me3), respectively[34,35]. PcG proteins contribute to the maintenance of the anterior-posterior (A-P) axis during mammalian embryogenesis through repression of Hox genes in the neural tube[36] as well as that of forebrain-related genes in the midbrain[37]. Moreover, PcG proteins participate in cell subtype specification in the CTX in a manner dependent on temporal codes[38–43]. However, it has remained unknown whether PcG proteins regulate D-V patterning of the mammalian CNS including the telencephalon. We now show that Ring1, an E3 ubiquitin ligase for H2AK119ub and essential component of PRC1[44,45], is required for the formation of the ventral telencephalon. Neural-specific ablation of Ring1B or of both Ring1A and Ring1B thus relieves the repression of genes encoding BMP and Wnt ligands in, as well as results in expansion of the expression of dorsal-specific genes into, the ventral telencephalon, whereas it attenuates expression of Shh and ventral-specific genes in this region. Importantly, the binding of Ring1B and deposition of H3K27me3 at certain BMP and Wnt ligand genes is found to be region-specific, namely, excluded from the dorsal midline. Furthermore, we find that forced activation of BMP or Wnt signaling suppresses Shh expression in explant cultures prepared from the embryonic telencephalon. Overall, our findings indicate that Ring1 establishes a permissive state for Shh expression in the ventral region of the telencephalon through suppression of BMP and Wnt signaling in this region. In other words, the establishment of the D-V morphogen pattern in the mouse telencephalon is dependent on region-specific suppression of unwanted positional cues by Ring1.

## Results

**Ring1 deletion causes morphological defects in telencephalon.**
To investigate the role of PcG proteins in the early stage of mouse telencephalic development, we deleted Ring1b with the use of the Sox1-Cre transgene, which confers the expression of Cre recombinase in the neuroepithelium from before E8.5[46]. We confirmed that expression of the Ring1B protein in the telencephalic wall was greatly reduced in Ring1b^flox/flox;Sox1-Cre (Ring1B KO) mice compared with that in Ring1b^flox/flox or Ring1b^flox/+ (control) mice, whereas the abundance of Ring1B in tissues outside of the telencephalic wall appeared unchanged in the Ring1B KO embryos at E9 (Supplementary Fig. 1h, i) and E10 (Supplementary Fig. 1a, c). Ring1B showed an even distribution along the D-V axis in both control and Ring1B KO embryos (Supplementary Fig. 1c). The expression of Ring1B in the telencephalic wall was also greatly reduced in mice lacking both Ring1B and its homolog Ring1A (Ring1a^−/−;Ring1b^flox/flox;Sox1-Cre, or Ring1A/B dKO, mice) compared with that in Ring1a^−/−;Ring1b^flox/flox or Ring1a^−/−; Ring1b^flox/+ (Ring1A KO) mice at E9 (Supplementary Fig. 1l, m) and E10 (Supplementary Fig. 1b, d). In addition, the level of H2AK119ub in the telencephalic wall was reduced in Ring1B KO mice and, to a greater extent, in Ring1A/B dKO mice at E9 (Supplementary Fig. 1j, k, n, o) and E10 (Fig. 1a–d). These results were thus consistent with the notion that Ring1A and Ring1B have overlapping roles in H2AK119 ubiquitylation and that Ring1B

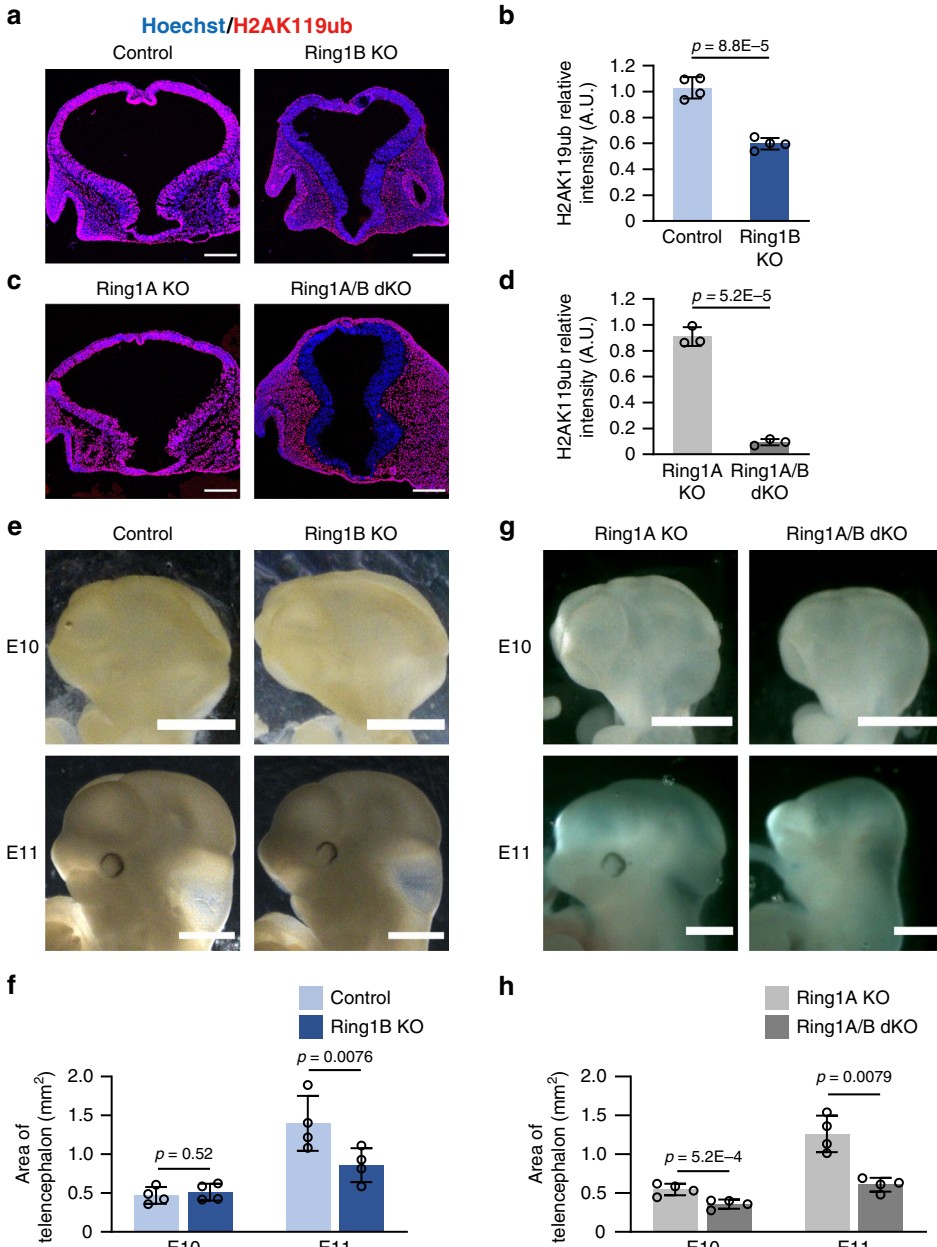

**Fig. 1 Deletion of *Ring1* in neural tissue reduces the amount of H2AK119ub and results in morphological defects in the early stage telencephalon.**
**a, c** Coronal sections of the brain of control (*Ring1b*^flox/flox^ or *Ring1b*^flox/+^) and Ring1B KO (*Ring1b*^flox/flox^;*Sox1-Cre*) mice (**a**) or of Ring1A KO (*Ring1a*^−/−^; *Ring1b*^flox/flox^ or *Ring1a*^−/−^;*Ring1b*^flox/+^) and Ring1A/B dKO (*Ring1a*^−/−^;*Ring1b*^flox/flox^;*Sox1-Cre*) mice (**c**) at E10 were subjected to immunohistofluorescence staining with antibodies to H2AK119ub. Nuclei were counterstained with Hoechst 33342. Scale bars, 200 μm. **b, d** The ratio of the average immunostaining intensity of H2AK119ub for the entire telencephalon to that for nonneural tissue adjacent to the ventral telencephalon was determined as relative intensity (A.U., arbitrary units) for images similar to those in **a** and **c**, respectively. Data are means ± s.d., *n* = 3 embryos of each genotype, two-tailed Student's unpaired *t* test. **e, g** Images of the brain of control and Ring1B KO mice (**e**) or of Ring1A KO and Ring1A/B dKO mice (**g**) at E10 and E11. Scale bars, 1 mm. **f, h** Quantification of the lateral projected area of the telencephalon at E10 and E11 for images similar to those in (**e**) and (**g**). Data are means ± s.d., averaged values for *n* = 4 litters (the number and individual values of each littermate are provided in Source Data File), two-tailed Student's paired *t* test. Source data are provided as a Source Data file.

makes a greater contribution to this modification than does Ring1A[34].

We found that *Ring1b* deletion with the use of *Sox1-Cre* resulted in a significant reduction in the size of the telencephalon at E11 (Fig. 1e, f). Deletion of *Ring1b* at E13.5 with the use of the *Nestin-CreERT2* transgene was previously found not to substantially affect the morphology or size of the telencephalon[38], indicating that Ring1B functions during the early stage of telencephalic development. The size of the telencephalon was

also reduced in Ring1A/B dKO mice compared with Ring1A KO mice (Fig. 1g, h). The number of cells positive for the cleaved form of caspase-3 (a marker for apoptosis) in the telencephalic wall was increased in Ring1B KO mice and, to a greater extent, in Ring1A/B dKO mice at E10 (Supplementary Fig. 1e–g), suggesting that Ring1 is necessary for the survival of telencephalic cells and that the reduction in telencephalic size induced by *Ring1* deletion is due, at least in part, to the aberrant induction of apoptosis.

**Ring1 deletion reduces the expression of ventral-specific TFs.** We next investigated whether *Ring1* deletion might affect the D-V axis of the telencephalon by determining the expression of region-specific TFs. We examined embryos mostly at E10 given the apparently normal size of the telencephalon in Ring1B KO mice at this stage (Fig. 1e, f). Nkx2.1 is a TF specifically expressed in the MGE[4], and we found that the expression of this protein was greatly diminished in Ring1B KO mice at E10 (Fig. 2a). The dorsal border of the Nkx2.1 expression domain was thus shifted ventrally and the abundance of Nkx2.1 within this domain was also reduced in response to *Ring1b* deletion (Fig. 2b, c). Loss of Nkx2.1 expression was also apparent in Ring1A/B dKO mice (Fig. 2d–f). Moreover, the deletion of *Ring1b* resulted in marked downregulation of *Nkx2.1* mRNA in CD133[+] NPCs isolated from the GE of the telencephalon at E11 (Fig. 2g). These results thus indicated that Ring1 is necessary for ventral expression of the MGE marker Nkx2.1.

We also examined the expression of Gsx2, which is highly enriched in the LGE and whose mRNA is present in both the LGE and MGE[8]. Immunostaining indeed revealed the expression of Gsx2 protein within nuclei of NPCs in the LGE of Ring1A KO mice at E10, whereas such expression was markedly attenuated in Ring1A/B dKO mice (Supplementary Fig. 2). Moreover, *Ring1b* deletion significantly reduced the level of *Gsx2* mRNA in CD133[+] NPCs isolated from the GE of the telencephalon at E11 (Fig. 2g). These results together indicated that Ring1 plays a role in the expression of ventral TFs in the ventral region of the telencephalon.

**Ring1 deletion expands CTX-specific TFs into ventral NPCs.** We then investigated whether the deletion of *Ring1* affects the expression of dorsal-specific TFs in the developing telencephalon. Pax6 contributes to the development of the CTX, with its expression being restricted to the dorsal pallium in mice[5]. However, we found that the expression of Pax6 extended to the ventral region of the telencephalon in Ring1B KO mice as well as in Ring1A/B dKO mice (Fig. 2h, i). Indeed, the D-V gradient of Pax6 expression was shallower in Ring1B KO mice and in Ring1A/B dKO mice compared with control mice and Ring1A KO mice, respectively (Fig. 2j, k). We also examined the role of Ring1 in the regulation of Emx1, another CTX-specific TF[3]. The deletion of *Ring1b* increased the amount of *Emx1* mRNA in CD133[+] NPCs isolated from the GE of the telencephalon at E11 (Fig. 2g). These results together indicated that Ring1 suppresses the expression of dorsal TFs in the ventral region of the telencephalon and thereby prevents the dorsalization of this region during the early stage of development.

**Ring1 deletion dorsalizes proneural gene expression.** Given that *Ring1* deletion appeared to induce dorsalization of the expression patterns of TFs related to NPC specification along the D-V axis, we examined the expression of proneural genes that contribute to region-specific neuronal differentiation. The basic helix-loop-helix proteins Neurog1 and Ascl1 are pallium- and subpallium-specific proneural factors, respectively[7,9]. There was thus little overlap of Neurog1 and Ascl1 expression at the pallium-subpallium boundary of control mice at E10 (Fig. 3a). However, deletion of *Ring1b* resulted in a ventral shift of the ventral border of Neurog1 expression and a marked overlap of Neurog1 expression with Ascl1 expression in the ventral region (Fig. 3a–d, Supplementary Fig. 3), again suggesting that loss of Ring1 induces dorsalization of the telencephalon. The deletion of *Ring1b* did not obviously shift the dorsal border of Ascl1 expression (Fig. 3c) but significantly reduced the level of Ascl1 within the LGE (Fig. 3d). In Ring1A/B dKO mice, the ventral border of the Neurog1[+]

region was also shifted ventrally (Fig. 3e, g), and the level of Ascl1 protein in the ventral region appeared to be reduced to a greater extent than in Ring1A KO (Fig. 3f). These results supported the notion that Ring1 plays a pivotal role in the establishment of ventral identity in the early stage of telencephalic development.

**Ring1 represses BMP and Wnt signaling in ventral telencephalon.** We next investigated whether *Ring1* deletion affects the gene expression profile of NPCs in the ventral telencephalon. Transcripts isolated from CD133[+] NPCs derived from the GE of the control or Ring1B KO telencephalon at E11 were subjected to RNA-sequencing (RNA-seq) analysis with the Quartz amplification method[47]. Differentially expressed genes were determined with the use of *edgeR* of the *R* package[48,49]. We identified more upregulated genes (953) than downregulated genes (238) in ventral NPCs from Ring1B KO mice compared with those from control mice (Fig. 4a–c), consistent with the general role of PcG proteins in gene repression. Importantly, the expression of genes for dorsal-specific TFs such as *Emx1* was upregulated, whereas that of genes for ventral-specific TFs such as *Nkx2.1* and *Olig2*[50,51] was downregulated, in the NPCs from Ring1B KO mice (Fig. 4b, c). These results thus confirmed the role of Ring1 in the suppression of the dorsalization of ventral NPCs.

To shed light on the mechanism by which Ring1 establishes (or maintains) ventral identity in ventral telencephalic NPCs, we performed KEGG (Kyoto Encyclopedia of Genes and Genomes) pathway analysis for both the upregulated and downregulated gene sets with the use of DAVID Bioinformatic Resources[52,53] (Fig. 4d, e). Among the genes whose expression was upregulated in NPCs of Ring1B KO mice, pathway analysis revealed an enrichment of categories such as pathways in cancer and extracellular matrix (ECM)–receptor interaction (Fig. 4d), with this enrichment being due, at least in part, to derepression of protocadherin-γ and collagen family genes (Supplementary Data 1). Furthermore, we found that Hippo, Wnt, and transforming growth factor-β (TGF-β) signaling pathways were also enriched among the genes whose expression was upregulated by *Ring1b* deletion (Fig. 4d). The upregulated genes categorized in the Hippo signaling pathway included genes related to BMP and Wnt signaling pathways (Supplementary Fig. 4b). Indeed, expression of the BMP and Wnt ligand genes *Bmp5*, *Bmp6*, *Bmp7*, *Wnt5b*, *Wnt7b*, and *Wnt8b* was increased by *Ring1b* deletion (Supplementary Fig. 4c). The RT-qPCR analysis also showed that the expression of *Bmp4* and *Id1*, which are major components of the BMP signaling pathway, as well as that of *Wnt7b*, *Wnt8b*, and *Axin2*, which are major players in Wnt signaling, were increased in Ring1B-null NPCs compared with control NPCs (Fig. 4f, g). These results together indicated that Ring1B suppresses BMP and Wnt signaling pathways in NPCs of the ventral telencephalon at E11.

We next investigated how Ring1 deletion affects the distribution of transcripts of *Wnt8b*, as well as those of *Axin2* as a monitor of Wnt signaling activity, with in situ hybridization (RNAscope) analysis[54]. The highest level of *Wnt8b* and that of *Axin2* expression are normally confined to the dorsal midline (presumptive cortical hem), with a lower level of expression also occurring in the pallium (presumptive neocortex) of the early stage telencephalon. However, the regions showing the greatest abundance of *Wnt8b* or *Axin2* mRNA at E10 were expanded in Ring1A/B dKO mice compared with Ring1A KO mice (Fig. 5a–g), suggesting that Ring1 suppresses the expression of *Wnt8b* and the activity of Wnt signaling outside of the dorsal midline. We also examined the distribution of *Bmp4* mRNA and Id1 protein, as a monitor of BMP signaling activity. The highest level of *Bmp4* and

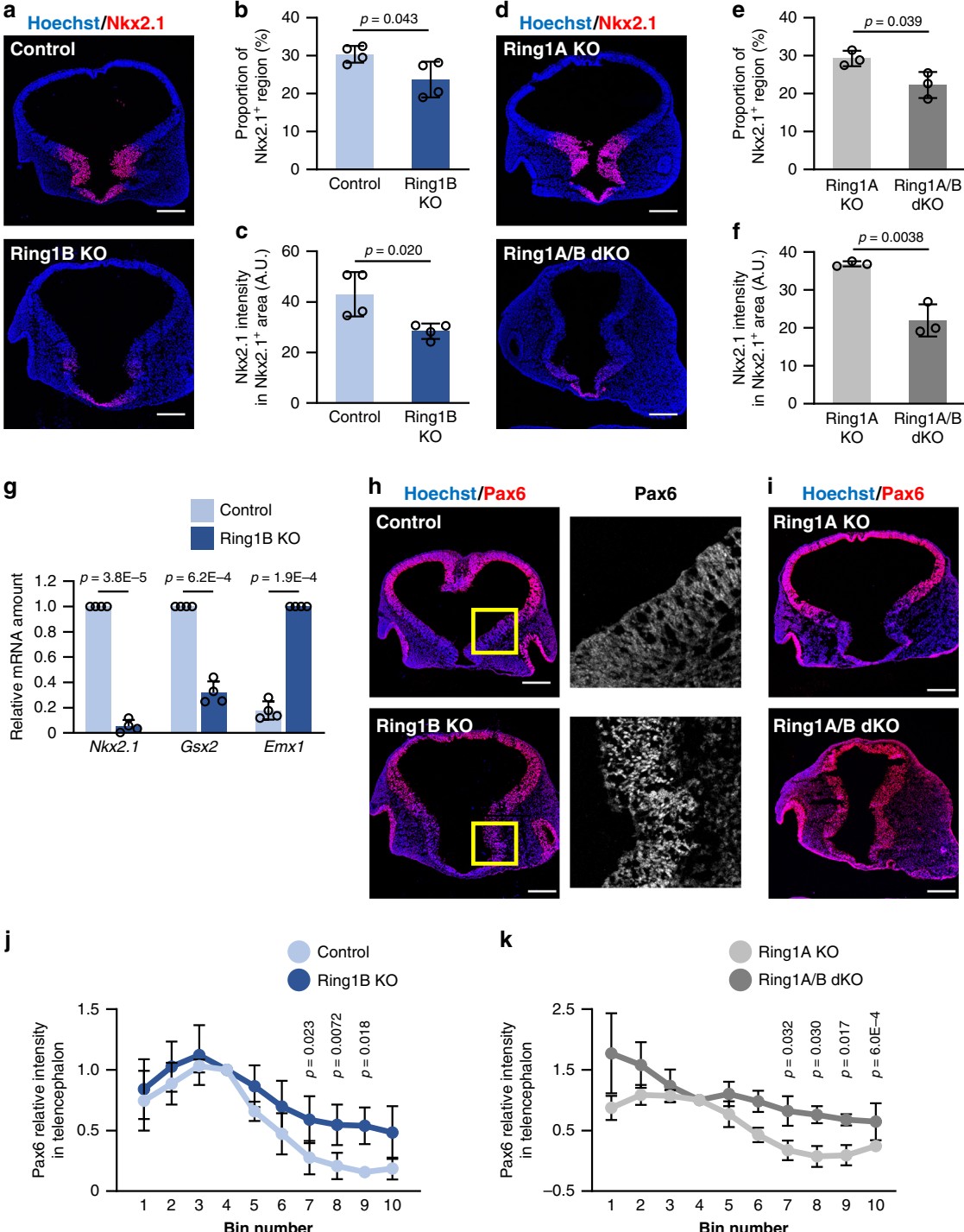

**Fig. 2 Deletion of *Ring1* confers a dorsal-like expression pattern of region-specific TFs in NPCs of the ventral telencephalon. a, d, h, i** Coronal sections of the brain of control and Ring1B KO mice (**a, h**) or of Ring1A KO and Ring1A/B dKO mice (**d, i**) at E10 were subjected to immunohistofluorescence staining with antibodies to Nkx2.1 (**a, d**) or to Pax6 (**h, i**). Nuclei were counterstained with Hoechst 33342. The regions indicated with yellow outlined squares (300 by 300 μm) in the left panels of (**h**) are shown at higher magnification in the right panels. Scale bars, 200 μm. **b, c, e, f** Quantification of immunostaining for Nkx2.1 in images similar to those in (**a**) and (**d**). The Nkx2.1+ perimeter length as a proportion of total perimeter length for the telencephalic wall (**b, e**) as well as the average Nkx2.1 immunostaining intensity (in arbitrary units, A.U.) within the Nkx2.1+ region (pixels) for each section (**c, f**) were determined. Data are means ± s. d., $n = 4$ (**b, c**) and $n = 3$ (**e, f**) embryos of each genotype, two-tailed Student's unpaired *t* test. **g** Reverse transcription (RT) and quantitative polymerase chain reaction (qPCR) analysis of relative *Nkx2.1*, *Gsx2*, or *Emx1* mRNA abundance (normalized by the amount of *Actb* mRNA) in CD133+ NPCs isolated from the GE of control or Ring1B KO mice at E11. Data are means ± s.d., averaged values for $n = 4$ litters (the number and individual values of each littermate are provided in Source Data File), two-tailed Student's paired *t* test. **j, k** Quantification of immunostaining intensity of Pax6 for images similar to those in (**h**) and (**I**), respectively. The telencephalic wall was divided into 10 bins, from 1 (dorsal) to 10 (ventral), and the average immunostaining intensity of Pax6 was determined in each bin and normalized by the average value for bin 4. Data are means ± s.d., $n = 4$ embryos from each genotype, two-tailed Student's paired *t* test between values in each bin. Source data are provided as a Source Data file.

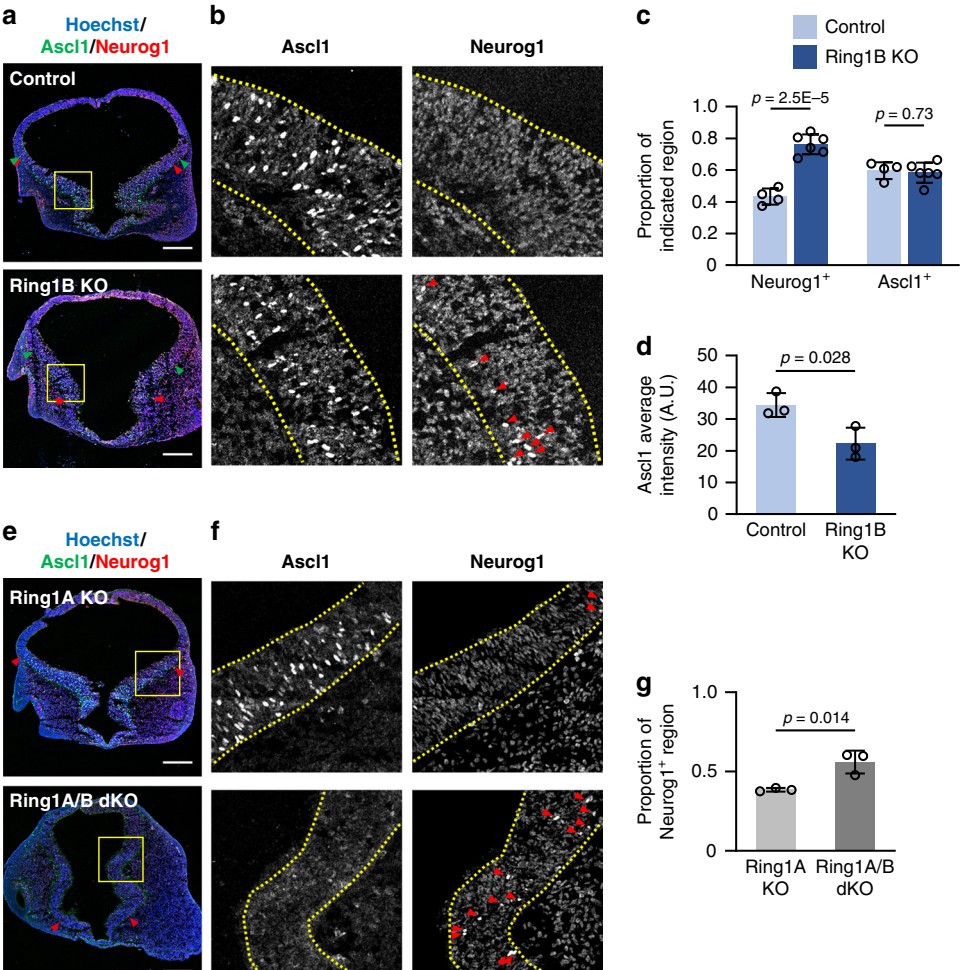

**Fig. 3 Deletion of *Ring1* confers dorsalized expression patterns of proneural genes in the ventral telencephalon. a, b, e, f** Coronal sections of the brain of control and Ring1B KO mice (**a**) or of Ring1A KO and Ring1A/B dKO mice (**e**) at E10 were subjected to immunohistofluorescence staining with antibodies to Neurog1 and to Ascl1. Nuclei were counterstained with Hoechst 33342. The regions indicated with yellow outlined squares (250 by 250 μm in (**a**), 300 by 300 μm in (**e**)) are shown at higher magnification in (**b**) and (**f**), respectively. Red arrowheads in ((**a**) and (**e**)) and green arrowheads in **a** represent the ventral and dorsal border of Neurog1⁺ and Ascl1⁺ regions, respectively. The telencephalic wall is demarcated by yellow dotted lines in (**b**) and (**f**). Red arrowheads in (**b**) and (**f**) indicate Neurog1⁺ cells. Scale bars, 200 μm. **c, d** Neurog1⁺ perimeter length or Ascl1⁺ perimeter length was determined as a proportion of total perimeter length for the telencephalic wall of control and Ring1B KO mice (**c**). The average of Ascl1 immunostaining intensity within Ascl1⁺ cells in the LGE was also determined for each section and then corrected for the intensity in the dorsal telencephalon (**d**). Data are means ± s.d., $n$ = 4 control and $n$ = 6 Ring1B KO mice (**c**) or $n$ = 3 control and Ring1B KO mice (**d**), two-tailed Student's unpaired *t* test. **g** Neurog1⁺ perimeter length of Ring1A KO and Ring1A/B dKO embryos was determined as in **c**. Data are means ± s.d., $n$ = 3 embryos of each genotype, two-tailed Student's unpaired *t* test. Source data are provided as a Source Data file.

that of Id1 expression are normally confined to the dorsal midline, whereas the expansion of these regions to the entire ventricular wall of the telencephalon was apparent in Ring1A/B dKO mice at E10 (Fig. 5h–n; also at E9 for Id1, Supplementary Fig. 5). These results suggested that Ring1 suppresses the expression of *Bmp4* and the activity of BMP signaling outside of the dorsal midline at the early stage of telencephalic development.

**Ring1 deletion reduces *Shh* signaling.** KEGG pathway analysis revealed an enrichment of genes related to the Hedgehog signaling pathway among the genes whose expression was downregulated in Ring1B-null NPCs (Fig. 4e). Consistent with this finding, the expression levels of genes related to Shh signaling—such as *Gli1*, *Gli2*, *Ptch1*, and *Ptch2*—were significantly lower in ventral NPCs from Ring1B KO mice compared with those from

control mice at E11 (Fig. 6a, b), suggesting that Ring1B is essential for activation of Shh signaling in ventral NPCs.

Given the attenuated expression of Shh target genes in Ring1B-null ventral NPCs, we examined whether the deletion of *Ring1b* might affect the expression of *Shh*. In situ hybridization analysis at E10 revealed that *Ring1b* deletion markedly reduced the abundance of *Shh* mRNA, which is normally found at the ventral midline (presumptive preoptic area) of the telencephalon at this stage (Fig. 6c, d). Furthermore, *Shh* expression was not apparent in the telencephalon of Ring1A/B dKO embryos (Fig. 6e). These results together indicated that Ring1 is required for expression of the gene for Shh, the major ventral morphogen, which might explain the dorsalization phenotype of the Ring1-deficient telencephalon.

It remained unclear, however, whether the downregulation of Shh signaling induced by *Ring1* deletion was due simply to the attenuation of *Shh* expression or was also due to an inability of NPCs to express target genes in response to Shh. We therefore

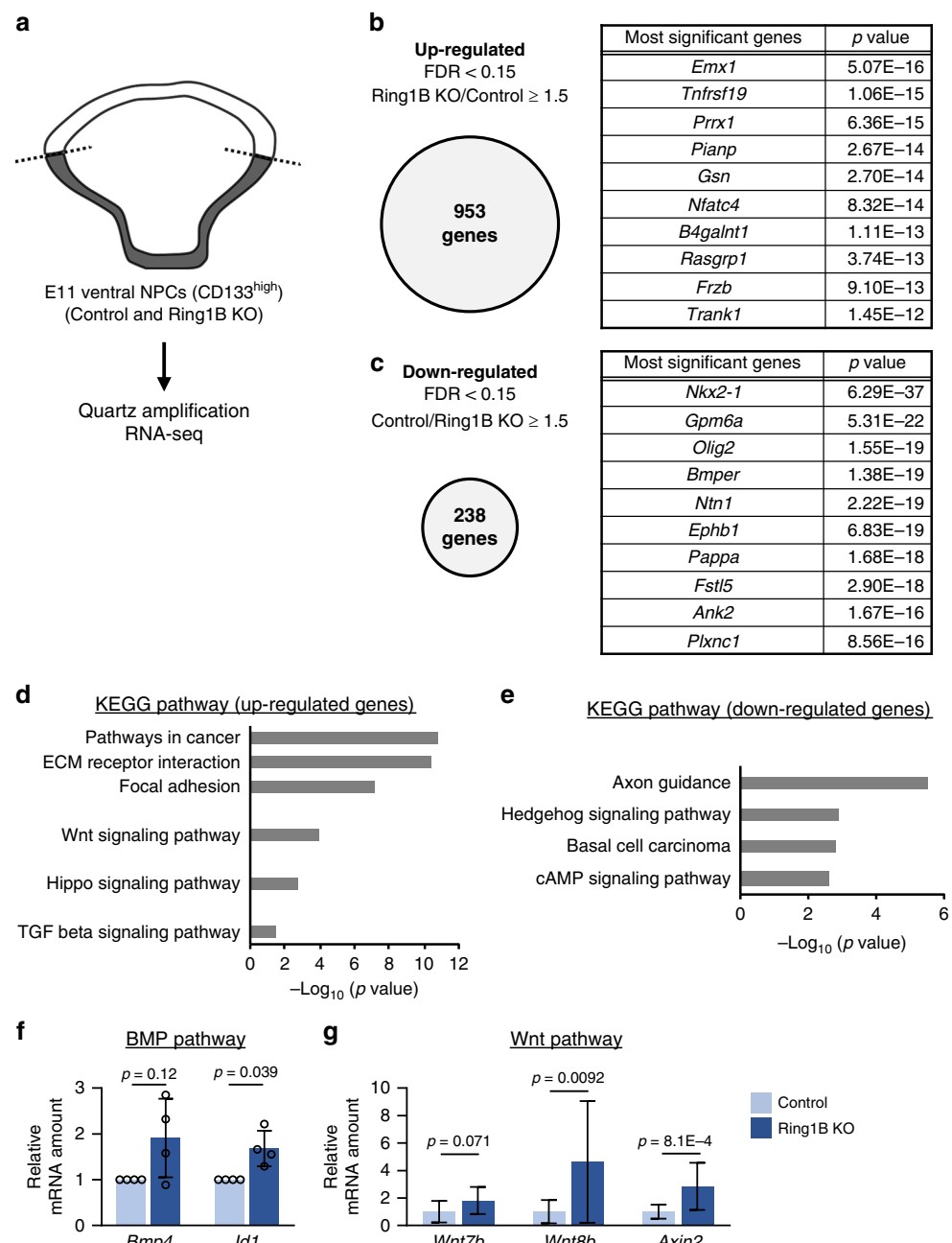

**Fig. 4 Genome-wide gene expression analysis of Ring1B-null NPCs derived from the ventral telencephalon. a** Genome-wide gene expression in NPCs isolated from the GE of control or Ring1B KO mice at E11 was analyzed by Quartz-seq analysis. A total of three samples prepared from one, one, or two embryos was analyzed for each genotype. **b**, **c** Genes whose expression was upregulated (**b**) or downregulated (**c**) in NPCs of Ring1B KO mice were defined as those whose Ring1B KO/control or control/Ring1B KO fold change, respectively, was ≥1.5 on average and ≥1.2 in each experiment, with a false discovery rate (FDR) of <0.15 (left panels). The genes with the 10 lowest $p$ values (determined with *edgeR*) in each category are also listed (right panels). **d**, **e** Enriched pathways among upregulated genes (**d**) and downregulated genes (**e**) were determined by KEGG pathway analysis. $p$ values were determined with DAVID Bioinformatic Resources. For the full list of differentially expressed genes and enriched pathways, see Supplementary Data 1. **f** RT-qPCR analysis of the relative abundance of *Bmp4* and *Id1* mRNAs (normalized by the amount of *Actb* mRNA) in NPCs of control or Ring1B KO mice at E11. Data are means ± s.d., averaged values for $n = 4$ l (the number and individual values of each littermate are provided in Source Data File), two-tailed Student's paired $t$ test. **g** RT-qPCR analysis of the relative abundance of *Wnt7b*, *Wnt8b*, and *Axin2* mRNAs (normalized by the amount of *Actb* mRNA) in NPCs of control or Ring1B KO mice at E11. Data are means ± s.d., $n = 13$–16 (15 for *Wnt7b*, 13 for *Wnt8b*, 16 for *Axin2*) control and $n = 5$ Ring1B KO mice from three litters, two-tailed Student's unpaired $t$ test. Source data are provided as a Source Data file.

prepared in vitro cultures of telencephalic NPCs at E10 and examined their competence for Shh signaling by the addition of a smoothened agonist (SAG). The extent of the induction of the Shh target genes *Gli1* and *Ptch1* were similar for NPCs isolated from Ring1B KO mice and from control mice (Fig. 7a, b),

suggesting that *Ring1b* deletion did not substantially affect the regulation of these Shh target genes and that the regulation of *Shh* expression is crucial for Ring1-dependent ventral identity.

How does Ring1 promote *Shh* expression? Given the general role of PcG proteins in gene repression, it was plausible that

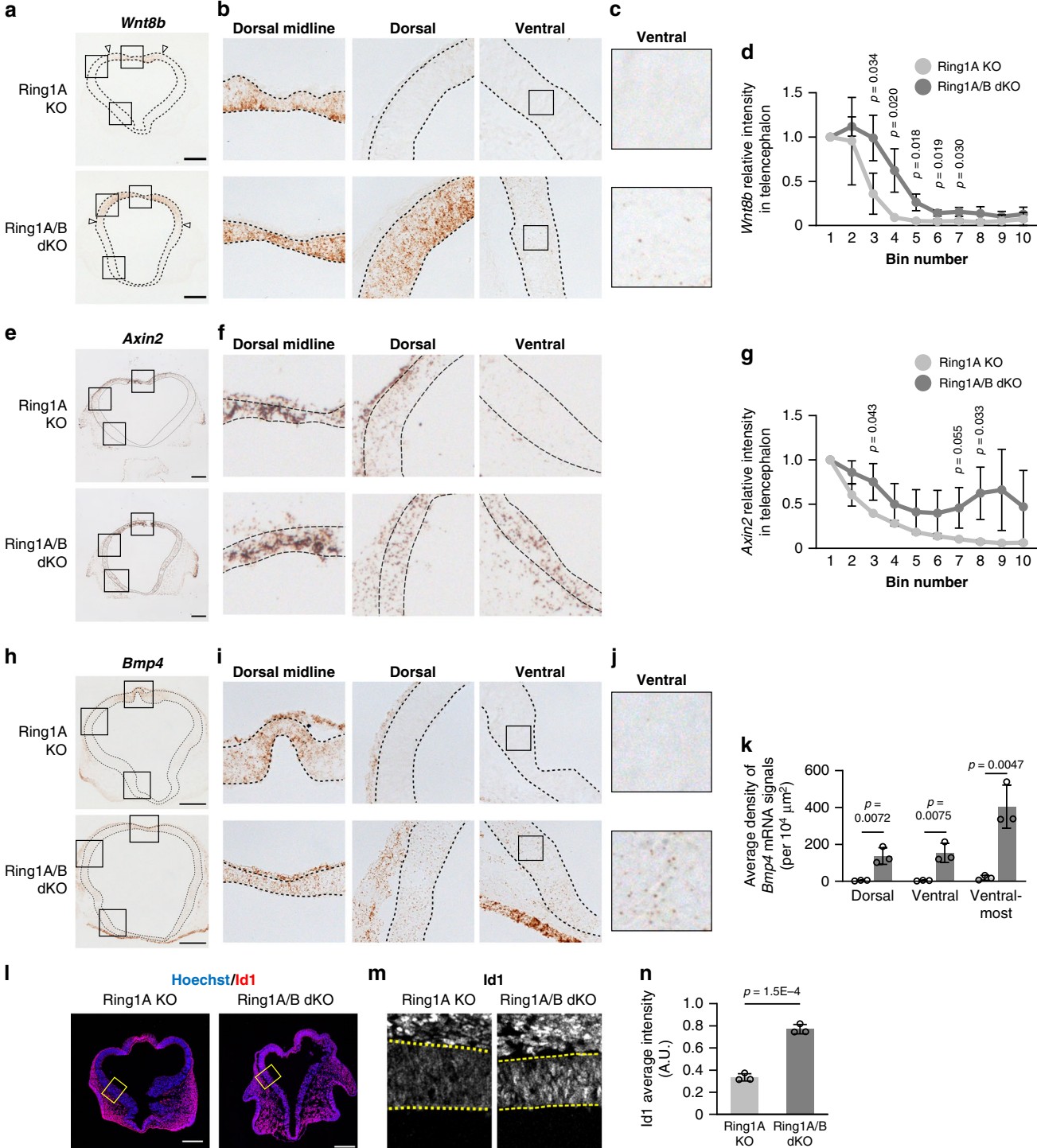

**Fig. 5 Deletion of *Ring1* activates BMP and Wnt signaling in the early stage telencephalon. a–c, e, f, h–j** Coronal sections of the brain of Ring1A KO or Ring1A/B dKO mice at E10 were subjected to in situ hybridization analysis of *Wnt8b* (**a–c**), *Axin2* (**e, f**), or *Bmp4* (**h–j**) mRNA with the use of RNAscope[54]. The regions indicated with outlined squares (200 by 200 μm in **a** and **h**, 300 by 300 μm in **e**, 40 by 40 μm in **b** and **i**) are shown at higher magnification in (**b**), (**f**), (**l**), (**c**), and (**j**), respectively. The telencephalic wall is demarcated by dotted lines. Arrowheads in (**a**) represent the ventral border of *Wnt8b*^high region. Scale bars, 200 μm. **d, g** Quantification of intensity of *Wnt8b* and *Axin2* mRNA for images similar to those in (**a**) and (**e**), respectively. The telencephalic wall was divided into 10 bins, from 1 (dorsal) to 10 (ventral), and the average intensity was determined in each bin and normalized by the average value for bin 1. Data are means ± s.d., n = 3 embryos of each genotype. **k** The average density of *Bmp4* signals (per $10^4$ μm$^2$) of the dorsal, ventral, and ventralmost regions of the Ring1A KO and Ring1A/B dKO mouse telencephalon was determined for images as in (**h**). Data are means ± s.d., n = 3 embryos of each genotype. **l, m** Coronal sections of the brain of Ring1A KO or Ring1A/B dKO mice at E10 were subjected to immunohistofluorescence staining with antibodies to Id1. The regions indicated with yellow outlined rectangles (150 by 200 μm) in (**l**) are shown at higher magnification in (**m**). Nuclei were counterstained with Hoechst 33342. The telencephalic wall is outlined by yellow dotted lines in (**m**). Scale bars, 200 μm. **n** The average intensity of Id1 signals in the ventral 90% of the telencephalic wall normalized by that for the dorsal 10% (dorsal midline) was determined from images similar to those in (**l**). Data are means ± s.d., n = 3 embryos of each genotype. Two-tailed Student's unpaired *t* test. Source data are provided as a Source Data file.

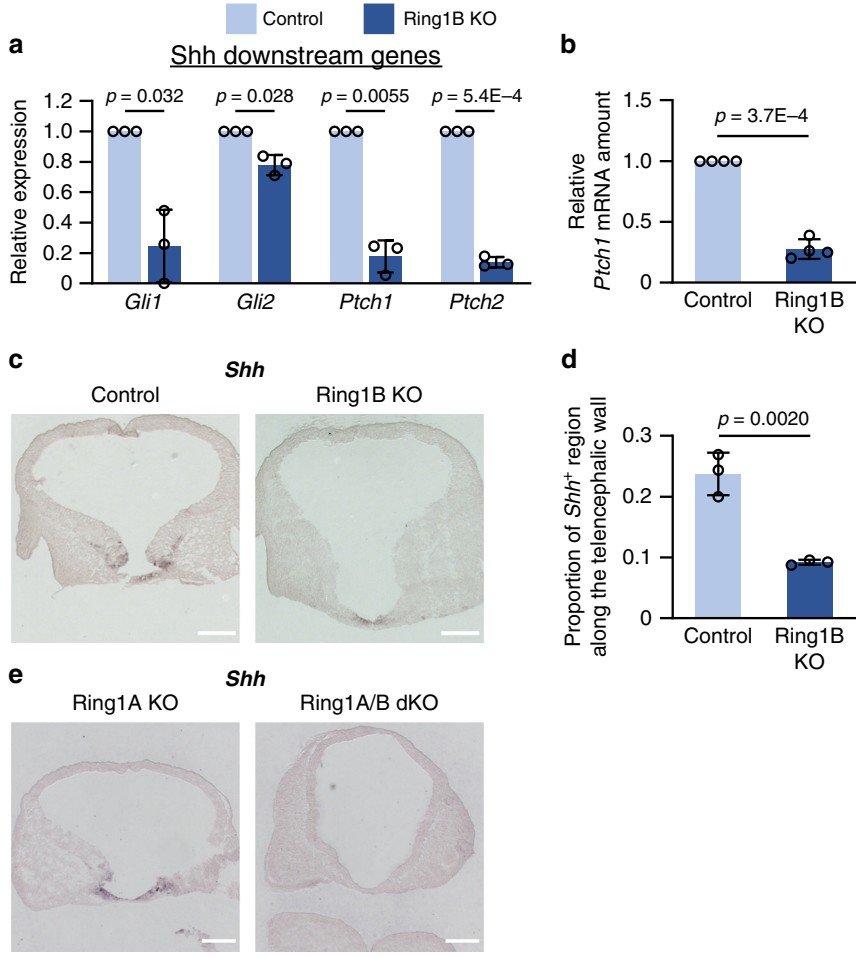

**Fig. 6 Deletion of *Ring1* attenuates the expression of *Shh* in the telencephalon. a** The RPKM (reads per kilobase of exon per million mapped reads) scores for *Gli1*, *Gli2*, *Ptch1*, and *Ptch2* in the RNA-seq analysis of NPCs from Ring1B KO embryos shown in Fig. 4 were normalized by those for the corresponding control sample in each experiment. Data are means ± s.d., $n = 3$ experiments. **b** RT-qPCR analysis of relative *Ptch1* mRNA abundance (normalized by the amount of *Actb* mRNA) in ventral NPCs of control or Ring1B KO mice at E11. Data are means ± s.d., averaged values for $n = 4$ litters (the number and individual values of each littermate are provided in Source Data File), two-tailed Student's paired *t* test. **c** Coronal sections of the brain of control or Ring1B KO mice at E10 were subjected to in situ hybridization analysis of *Shh* mRNA. Scale bars, 200 μm. **d** The *Shh* mRNA⁺ perimeter length as a proportion of total perimeter length for the telencephalic wall determined from sections similar to those in (**c**). Data are means ± s.d., $n = 3$ embryos of each genotype, two-tailed Student's unpaired *t* test. **e** Coronal sections of the brain of Ring1A KO or Ring1A/B dKO mice at E10 were subjected to in situ hybridization analysis of *Shh* mRNA. Scale bars, 200 μm. Source data are provided as a Source Data file.

Ring1 indirectly increases *Shh* expression through repression of genes whose products inhibit *Shh* expression. Implantation of beads soaked with recombinant BMP in the anterior neuropore of chick embryos was previously shown to inhibit *Shh* expression[33]. Furthermore, forced activation of canonical Wnt signaling by expression of a stabilized form of β-catenin was found to result in repression of ventral marker genes such as *Nkx2.1* in the mouse subpallium[22]. It was therefore possible that activation of BMP and Wnt signaling pathways might account for the downregulation of *Shh* expression in Ring1-deficient mice, although it remained unclear whether Wnt signaling is able to regulate the level of *Shh* expression. We therefore examined whether activation of Wnt signaling can reduce the level of *Shh* mRNA in collaboration with BMP signaling in dissociated (monolayer) and explant cultures prepared from the telencephalon of wild-type (WT) mice at E9. The addition of an activator of canonical Wnt signaling (the glycogen synthase kinase–3 inhibitor CHIR-99021) indeed significantly reduced the abundance of *Shh* mRNA in both the explant and dissociated cultures (Fig. 7c–f) under conditions in which it induced *Axin2* expression (Supplementary

Fig. 6). Moreover, exposure to both BMP4 and CHIR-99021 tended to have a greater effect on the amount of *Shh* mRNA in the dissociated culture than did either agent alone (Fig. 7f). The activation of both BMP and Wnt signaling may therefore cooperate to suppress *Shh* expression at this early stage, consistent with the notion that the Ring1-dependent establishment of ventral identity is mediated by suppression of such signaling.

However, it was also possible that Ring1 somehow directly activates the expression of *Shh* and thereby suppresses the expression of BMP and Wnt ligands. We therefore examined whether suppression of Shh signaling might increase the expression of genes related to BMP or Wnt signaling in monolayer cultures of NPCs. The addition of an inhibitor of Shh signaling (the Smoothened inhibitor cyclopamine) did not significantly affect the expression of BMP or Wnt ligand genes (*Bmp4*, *Bmp7*, *Wnt7b*, and *Wnt8b*) or that of downstream genes (*Id1* and *Axin2*) under conditions in which it reduced the amount of *Gli1* mRNA (Fig. 7g–i). These results supported the notion that Ring1 suppresses BMP and Wnt ligand gene expression independently of the regulation of *Shh* expression.

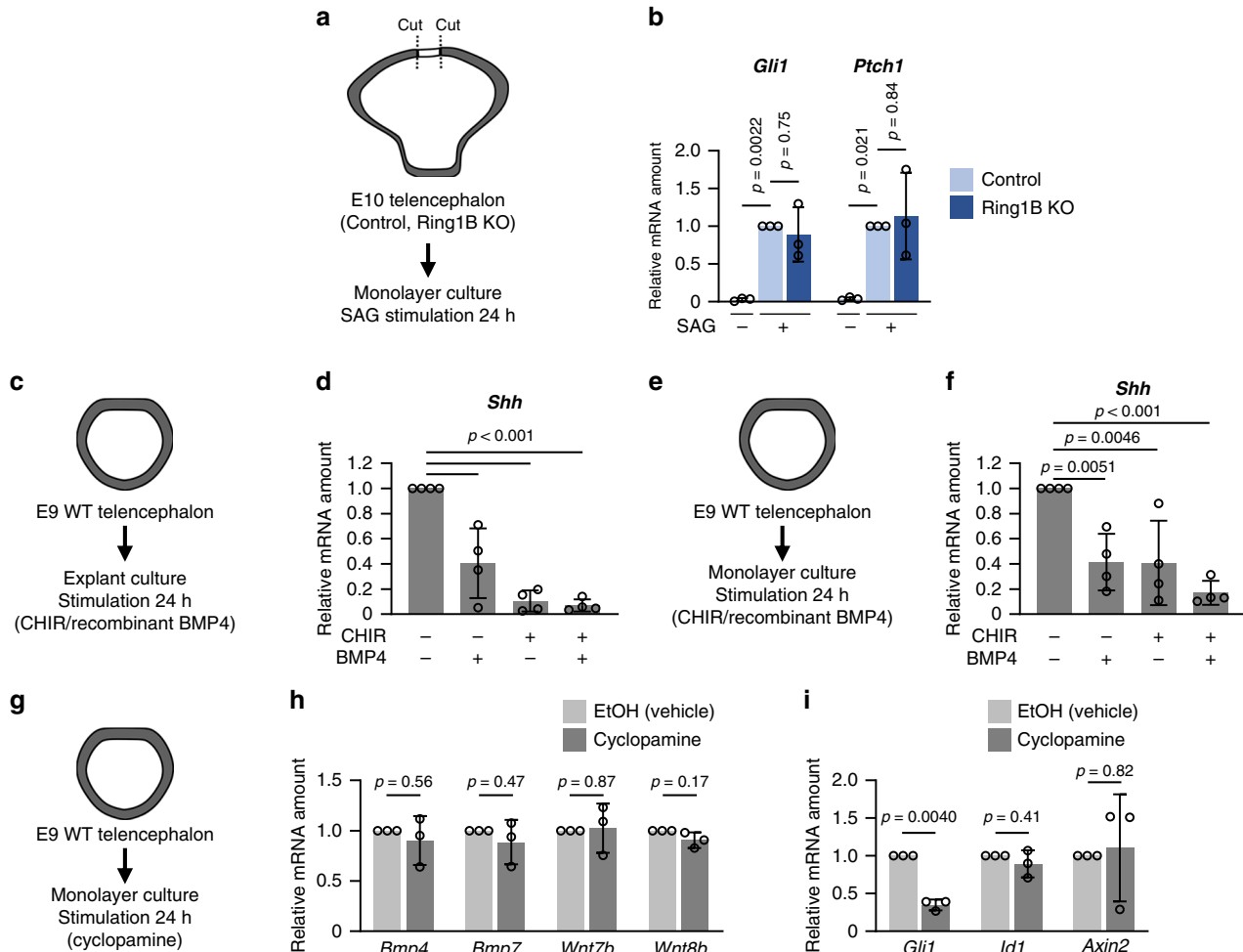

**Fig. 7 Forced activation of BMP and Wnt signaling inhibits *Shh* expression. a** Cells from the telencephalon (outside of the dorsal midline) of control or Ring1B KO mice at E10 were maintained in monolayer culture for 6 h and then exposed for 24 h to 2 μM SAG or dimethyl sulfoxide vehicle. **b** RT-qPCR analysis of relative *Gli1* and *Ptch1* mRNA abundance (normalized by the amount of *Actb* mRNA) in NPCs treated as in (**a**). Data are means ± s.d., $n = 3$ independent experiments, one-way ANOVA followed by Dunnett's multiple-comparison test. **c, e** The telencephalon of WT (ICR) mice at E9 was maintained in explant (**c**) or monolayer (**e**) culture for 6 h before exposure for an additional 24 h to BMP4 (50 ng/ml) or 5 μM CHIR-99021. **d, f** RT-qPCR analysis of relative *Shh* mRNA abundance (normalized by the amount of *Actb* mRNA) in NPCs treated as in (**c**) and (**e**), respectively. Data are means ± s.d., $n = 4$ independent experiments, one-way ANOVA followed by Dunnett's multiple-comparison test. **g** The telencephalon of WT (ICR) mice at E9 was maintained in monolayer culture for 6 h before exposure for an additional 24 h to 1 μM cyclopamine or ethanol (EtOH) vehicle. **h, i** RT-qPCR analysis of relative *Bmp4*, *Bmp7*, *Wnt7b*, or *Wnt8b* (**h**) or *Gli1*, *Id1*, or *Axin2* (**i**) mRNA abundance (normalized by the amount of *Actb* mRNA) in NPCs treated as in (**g**). Data are means ± s.d., $n = 3$ independent experiments, two-tailed Student's paired *t* test. Source data are provided as a Source Data file.

**Direct and region-specific regulation of BMP and Wnt by PcG.** Given the dysregulation of BMP and Wnt ligand gene expression induced by *Ring1* deletion, we next examined whether these genes are direct targets of PcG proteins with the use of chromatin immunoprecipitation (ChIP)–qPCR assays for H3K27me3, a histone modification catalyzed by PRC2, and for Ring1B with the telencephalon of WT mice at E9 (Fig. 8a). We indeed detected significant deposition of H3K27me3 at the promoters of *Bmp4*, *Bmp7*, *Wnt7b*, and *Wnt8b* at levels similar to those apparent at the promoters of *Hoxa1* and *Hoxd3* (positive controls) (Fig. 8b). We also detected significant binding of Ring1B at the promoters of *Bmp4* and *Wnt8b* (Fig. 8c). These results suggested that PcG proteins directly regulate the expression of these Bmp and Wnt ligand genes in the early stage telencephalon. Of note, we found that, with the exception of *Bmp7*, *Ring1b* deletion with *Sox1-Cre* did not significantly reduce the levels of H3K27me3 apparent at these loci at E10 (Supplementary Fig. 7), suggesting that H3K27me3 deposition alone is not sufficient for the suppression of these *Bmp* and *Wnt* genes in the absence of Ring1B.

We then addressed whether PcG proteins regulate BMP and Wnt ligand gene expression in NPCs in a region-specific manner along the D-V axis. We manually dissected the mouse telencephalon at E11 into the dorsal midline (DM), CTX, and ventral telencephalon (V), isolated CD133[+] NPCs from these regions, and performed cleavage under targets and tagmentation (CUT&Tag)[55] and RT-qPCR analyses (Fig. 9a). We confirmed that the dissection procedure effectively separated the DM, CTX, and V regions by examination of the expression of corresponding marker genes (*Msx1*, *Bmp4*, and *Wnt8b* for DM; *Foxg1* for CTX and V; *Emx1* for CTX; and *Gsx2* and *Nkx2.1* for V) (Supplementary Fig. 8a). We adopted CUT&Tag for detection of H3K27me3 deposition and Ring1B binding in these experiments given its higher sensitivity compared with ChIP-seq. We indeed detected the deposition of H3K27me3 and Ring1B at the promoters of *Bmp7* and *Wnt7b* with CUT&Tag even with a smaller number of cells than that used for ChIP-qPCR (Fig. 9d, e; Supplementary Fig. 8f, g). We confirmed the deposition of H3K27me3 and Ring1B binding at *Hoxa1* and *Hoxd3*, examined

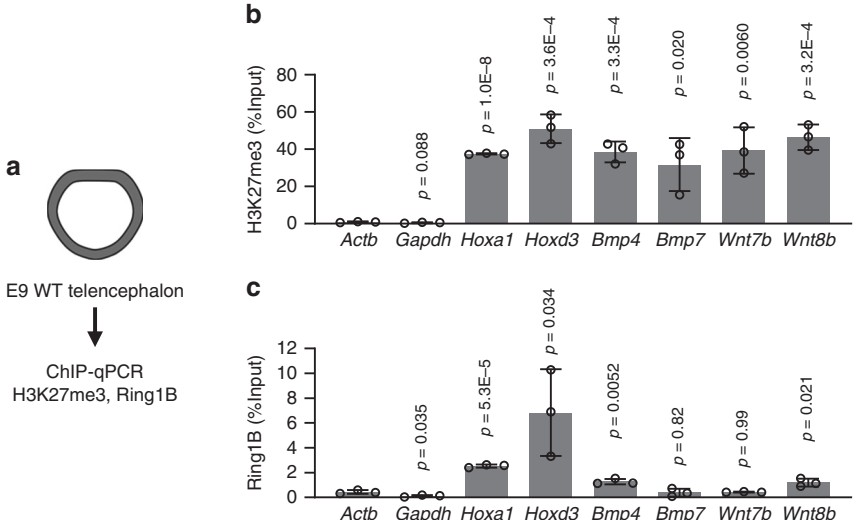

**Fig. 8 H3K27me3 deposition and Ring1B binding at BMP and Wnt ligand gene loci in early stage telencephalic NPCs. a** The telencephalon was isolated from WT (ICR) mice at E9 for ChIP-qPCR analysis with antibodies to H3K27me3 or to Ring1B. **b, c** ChIP-qPCR analysis of H3K27me3 deposition (**b**) and Ring1B binding (**c**) at the indicated promoters as in (**a**). *Hoxa1* and *Hoxd3* were examined as positive controls, and *Actb* and *Gapdh* as negative controls. Data are expressed as percentage input and are means ± s.d., $n = 3$ independent experiments, two-tailed Student's unpaired $t$ test versus the corresponding value for *Actb*. Source data are provided as a Source Data file.

as positive controls, with no differences apparent among DM, CTX, and V regions (Fig. 9d, e; Supplementary Fig. 8d, e). Conversely, no substantial deposition of H3K27me3 or Ring1B binding was apparent at the negative control genes *Actb* and *Gapdh* (Fig. 9d, e; Supplementary Fig. 8b, c). However, the level of H3K27me3 at the promoter as well as that of Ring1B at the peak in the gene body (Ring1B summit) of *Bmp4* and *Wnt8b* were significantly lower or tended to be lower in the DM than in the CTX or V regions (Fig. 9b–e). A similar pattern was apparent for *Bmp7* and *Wnt7b*, which manifested significantly higher levels of Ring1B in the V than in the DM region (Fig. 9e). Together with our finding that Ring1 deletion preferentially derepressed *Bmp4* and *Wnt8b* expression outside of the DM (Fig. 5), these results thus indicated that PcG proteins suppress the expression of BMP and Wnt ligand genes in the telencephalon in a region-specific manner, with this suppression being greater in the ventral region than in the dorsal region, and that this region specificity might underlie the establishment of the D-V axis.

## Discussion

Morphogenetic signals and their downstream TFs determine regional identity along the D-V axis in the developing telencephalon. Mutual inhibition between dorsal and ventral TFs plays a pivotal role in the segregation and maintenance of regional identity, but the mechanisms that underlie the initial regional confinement of morphogen expression have remained largely unknown. We have now found that Ring1A and Ring1B, core components of PRC1, play an essential role in the establishment of the spatial expression patterns of morphogens along the D-V axis and in consequent regionalization of the telencephalon at the early stage of mouse development. Our results also indicate that Ring1B and H3K27me3 are associated with genes encoding dorsal morphogens—such as *Bmp4*, *Bmp7*, *Wnt7b*, and *Wnt8b*—selectively in NPCs localized outside of the dorsal midline, with this specificity possibly accounting for restriction of the expression of these genes to the dorsal region. Together with the observations that ectopic activation of BMP signaling[33] or Wnt signaling (this study) is able to suppress the expression of the gene for the ventral morphogen Shh in the developing telencephalon, these results suggest that

Ring1 suppresses the expression of dorsal morphogens in the ventral telencephalon and thereby generates a permissive state for *Shh* expression, which is essential for the establishment of ventral identity[25,26,30,56] (Fig. 10). Our results thus unveil an epigenetic foundation of initial D-V patterning of morphogens in the mouse telencephalon.

The deletion of *Ring1* not only induced the dorsalization of NPC identity but also resulted in aberrant expression patterns of proneural genes. Ascl1 and Neurog1 are expressed in a mutually exclusive manner in WT embryos, in part as a result of the repression of *Ascl1* expression by Neurog1 and Neurog2[9]. However, we found that *Ring1b* deletion resulted in a marked increase in the number of cells positive for both Neurog1 and Ascl1 (Supplementary Fig. 3). Given that H3K27me3 and Ring1B are deposited at the promoters of *Neurog1* and *Ascl1* in early stage NPCs[38,40] (Supplementary Fig. 9), PcG proteins may contribute to the precision of D-V patterning by establishing the mutually exclusive inhibition of *Neurog1* and *Ascl1* expression and thereby regulating the segregation of neurogenic properties between NPCs.

Deletion of *Ring1b* with the use of the *Nestin-CreERT2* transgene, which confers Cre expression in the entire CNS at E13.5[38], or deletion of the gene for the histone methyltransferase Ezh2 with the *Emx1-Cre* transgene, which is expressed in the dorsal telencephalon from E10.5[41,57], has been shown to induce neurogenesis through derepression of neurogenic genes. With the use of the *Sox1-Cre* transgene, which is expressed in the neuroepithelium from before E8.5[46], we have now examined the role of Ring1 in the early stage of telencephalic development, before the onset of the neurogenic phase. During this early stage (for example, at E9), we did not detect promotion of neurogenesis, as determined by the expression of the neuronal marker βIII-tubulin, in response to *Ring1b* deletion (Supplementary Fig. 10), suggesting that a Ring1-independent mechanism is responsible for the suppression of neurogenesis at this time. Of interest, deletion of *Ring1b* with the use of the *Foxg1-IRES-Cre* transgene, which is expressed in the entire telencephalon from ~E9.0[58], did not appear to induce dorsalization of the ventral telencephalon (Supplementary Fig. 11), suggesting that Ring1-mediated D-V patterning of the telencephalon takes place only during the early

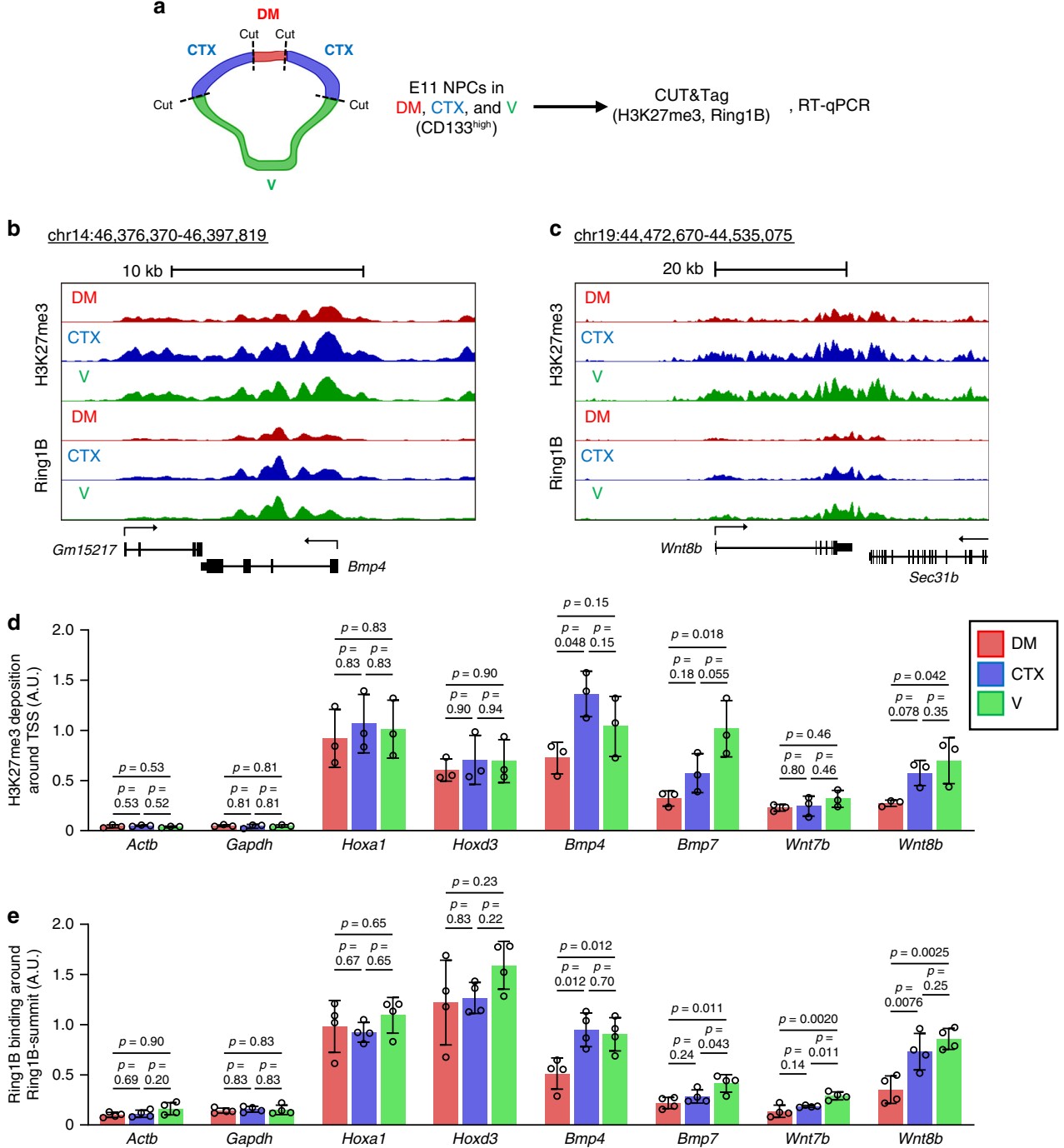

**Fig. 9 Telencephalic region-specific H3K27me3 deposition and Ring1B binding at BMP and Wnt ligand gene loci in early stage NPCs. a** CD133[+] NPCs isolated from the dorsal midline (DM), CTX, and ventral (V) regions of the telencephalon from WT (ICR) embryos at E11 were subjected to CUT&Tag analysis with antibodies to H3K27me3 or to Ring1B. Biological triplicates (for H3K27me3) or quadruplicates (for Ring1B) whose quality of regional dissection was confirmed by RT-qPCR analysis of region-specific gene expression (Supplementary Fig. 8a) were analyzed. **b**, **c** Averages of normalized CUT&Tag signals around *Bmp4* (**b**) or *Wnt8b* (**c**) in three (H3K27me3) or four (Ring1B) independent experiments are visualized in the UCSC genome browser. The RefSeq gene models are shown at the bottom of each panel. **d**, **e** Averages of normalized CUT&Tag signals for H3K27me3 around (±1 kbp) the transcription start site (TSS) (**d**) and Ring1B around (±1 kbp) the Ring1B summit (determined as the site in the gene body with the highest Ring1B signals across all samples) (**e**) of the indicated genes. Data are means ± s.d. for biological triplicates (**d**) or quadruplicates (**e**), one-way ANOVA followed by the Benjamini–Hochberg multiple-comparison test. Source data are provided as a Source Data file.

stage of development (earlier than E9.0), although the mechanism underlying this temporal restriction remains unclear.

BMP signaling has been shown to be important for the establishment of the dorsal midline and its activity to be confined to this region[18,59]. Our results now suggest that Ring1 mediates

suppression of BMP signaling outside of the dorsal midline and thereby sets up a permissive state for *Shh* expression. Given that the suppression of BMP signaling is necessary for neural induction of ectoderm[60], its onset may occur before the formation of the prospective forebrain. BMP signaling–related targets of PcG

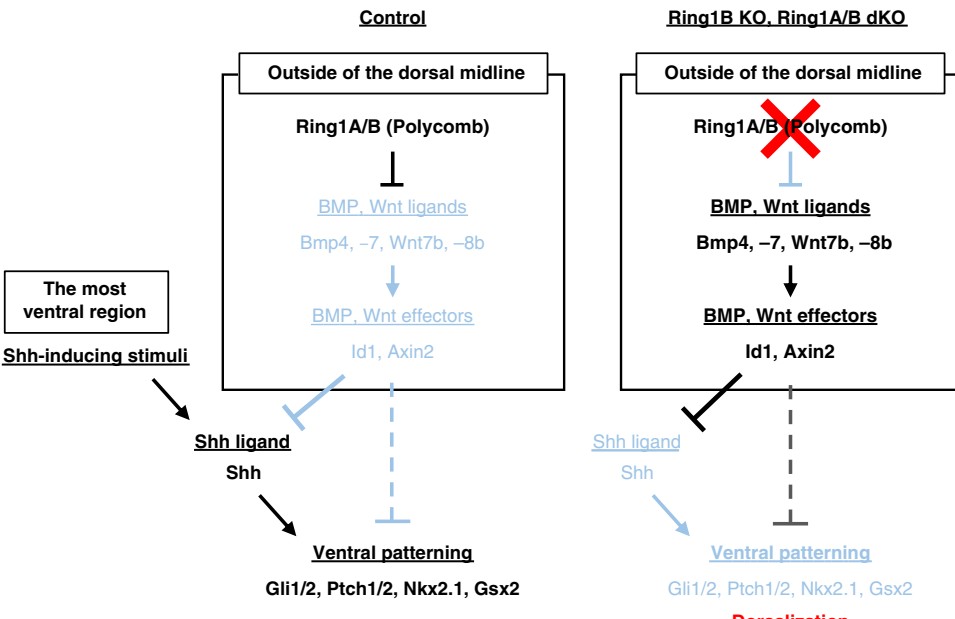

**Fig. 10 Model for Ring1-mediated ventral specification in the early stage mouse telencephalon.** Ring1 suppresses BMP and Wnt signaling pathways outside of the dorsal midline at the early stage of mouse telencephalic development and thereby confers a permissive state for *Shh* expression. Shh-inducing signals at the most ventral portion of the telencephalon can thus induce *Shh* expression in this region, resulting in ventral patterning of the telencephalon. By contrast, Ring1 ablation results in the ectopic activation of BMP and Wnt signaling outside of the dorsal midline and thereby suppresses Shh-mediated ventral patterning of the telencephalon.

proteins identified in embryonic stem cells may be involved in this early process[61]. The observed increase in Id1 expression outside of the dorsal midline in response to *Ring1* deletion from before E10 suggests that BMP signaling remains suppressed in this region but becomes activated at the dorsal midline, although the mechanisms underlying this temporal and spatial difference remain unknown.

A key related (and fundamental) question is how PcG proteins repress specific genes in specific regions at specific times. Given that we found that Ring1B is evenly expressed along the D-V axis at E10, the regional specificity of its action appears to be due to target-specific recruitment (or dissociation) of Ring1B. Of note, deletion of *Ezh2* in the dorsal midbrain with the use of the *Wnt1-Cre* transgene was previously shown to result in inhibition of Wnt signaling and to promote telencephalic identity (that is, rostralization) at ~E11.5[37], in contrast to our finding that *Ring1* deletion activates Wnt signaling in the early stage telencephalon. This previous study also showed that *Ezh2* deletion increased the expression of *Wif1* and *Dkk2*, both of which encode inhibitors of Wnt signaling, and that H3K27me3 was deposited at these gene loci in WT embryos, suggesting that PcG proteins contribute to Wnt activation by repressing these Wnt inhibitor genes in the dorsal midbrain. Mechanisms by which recruitment of PcG proteins is regulated in a tissue-, cell type-, or stage-specific manner warrant clarification in future studies.

The classical model of hierarchical PcG protein action proposes that PRC2-mediated H3K27me3 deposition triggers the recruitment of PRC1 (Ring1) to target loci[35,62]. However, recent studies have shown that PRC1 can also recruit PRC2 to their target loci[63]. The role of PRC2 in PcG protein-mediated gene repression is unclear—in particular, given that PRC2 components appear to be dispensable for target gene repression in embryonic stem cells[64] and that Ring1B inactivation results in derepression of target genes despite only a partial reduction in H3K27me3 levels in differentiating embryonic stem cells[65].

Our results now show that *Ring1b* deletion relieved the suppression of dorsal signaling in the ventral telencephalon in the absence of a significant reduction in H3K27me3 deposition at target gene loci, supporting the notion that H3K27me3 deposition alone is not sufficient for target gene repression also in an in vivo (developmental) context.

The robust maintenance of the A-P axis through suppression of *Hox* gene expression by PcG proteins has been well established from flies to mammals[66]. We now propose that PcG proteins also play an essential role in formation of the D-V axis in the early stage of mouse telencephalic development. Our study thus sheds light on the role of chromatin-level regulation in the regionalization of the brain that is dependent on developmental genes that are not necessarily clustered like *Hox* genes.

## Methods

**Animals.** *Ring1b*^flox/flox or *Ring1a*^−/−;*Ring1b*^flox/flox mice[67,68] were crossed with *Sox1-Cre* transgenic mice[46] or *Foxg1-IRES-Cre* transgenic mice[58]. Jcl:ICR (CLEA Japan) or Slc:ICR (SLC Japan) mice were studied as WT animals. All mice were maintained in a temperature- and relative humidity controlled (23° ± 3 °C and 50 ± 15%, respectively) environment with a normal 12-h-light, 12-h-dark cycle. They were housed two to six per sterile cage (Innocage, Innovive; or Micro BARRIER Systems, Edstrom Japan) with chips (PALSOFT, Oriental Yeast; or PaperClean, SLC Japan), and with irradiated food (CE-2, CLEA Japan) and filtered water available ad libitum. Mouse embryos were isolated at various ages, with E0.5 being considered the time of vaginal plug appearance. All animals were maintained and studied according to protocols approved by the Animal Care and Use Committee of The University of Tokyo.

**Plasmid constructs.** A pBluescript SK(-) vector encoding mouse *Shh* was kindly provided by D. Kawaguchi (The University of Tokyo). An amplified sequence is presented in Supplementary Table 3.

**Fixation and cryosection of embryos.** Isolated mouse embryos were fixed for 3 h or overnight with 4% paraformaldehyde (PFA) in phosphate-buffered saline (PBS). After washed with PBS, the embryos were incubated with 10%, 20%, and 30% sucrose in PBS and embedded with O.C.T. compound (Sakura Finetek Japan)

at −80 °C. Coronal sections of embedded embryos were prepared by Cryostar HM560 or NX70 (Thermo Fisher).

**Immunohistofluorescence analysis**. Immunohistofluorescence staining was performed as described[39] with some modifications. Coronal sections of embryos prepared from *Sox1-Cre* mice (only in the case of Gsx2 immunostaining, Supplementary Fig. 2) and *Foxg1-IRES-Cre* mice (Supplementary Fig. 11) were subjected to antigen retrieval with Target Retrieval Solution (Agilent) at 105 °C for 10 min. The sections were washed with tris-buffered saline containing 0.1% Triton-X100 (TBST) for 20 min and exposed to a blocking buffer (TBST containing 3% bovine serum albumin (BSA)) for 1 h. The sections were then incubated with primary antibodies solved in the blocking buffer at 4 °C for overnight, washed with TBST twice, incubated with secondary antibodies solved in the blocking buffer at room temperature for 1 h washed with TBST three times and mounted in Mowiol (Calbiochem). Images were acquired with a laser-scanning confocal microscope (TSC-SP5, Leica) and analyzed with ImageJ software (NIH). For quantification of protein abundance, the average value of multiple sections of the telencephalon along the rostrocaudal axis was determined for each embryo. Antibodies are listed in Supplementary Table 4.

**Isolation of NPCs by FACS**. The telencephalon was dissected into the indicated regions and subjected to enzymatic digestion with a papain-based solution (FUJIFILM Wako Chemicals). Cell suspensions in 0.2% BSA/PBS were stained with allophycocyanin-conjugated antibodies to CD133 (141210, BioLegend, Supplementary Table 5) at a dilution of 1:400 and were then subjected to fluorescence-activated cell sorting (FACS) with the FACSAria instrument (Becton Dickinson). CD133$^+$ NPCs were isolated as the top 50% of allophycocyanin-positive cells. Gating strategies are presented in Supplementary Fig. 4a.

**Quartz-seq analysis**. Both cDNA synthesis and amplification were performed as described[47] with some modifications. Two thousand cells were collected in a Buffer RLT (Qiagen) with FACS and then the RNA was purified with 20.5 μl magnetic beads (RNAClean XP; Beckman Coulter) in the presence of an RNase inhibitor (RNasin plus; Promega). The beads were then washed with 80% ethanol three times at room temperature and briefly dried (1–5 min air dry) and mixed with 12 μl Priming buffer (0.83× PCR buffer (TAKARA), 83.3 μM dNTPs, 69.4 nM reverse transcription (RT)-primer). After priming at 70 °C for 90 s and 35 °C at 15 s, the sample was kept on ice and mixed with 8 μl RT buffer (1× PCR buffer, 12.5 mM dithiothreitol, SuperScript III (Invitrogen)). After RT reaction at 35 °C for 5 min, 45 °C for 20 min and 70 °C for 10 min, cDNAs were purified with 36 μl magnetic beads (AMpure XP; Beckman Coulter) and eluted in 6 μl Exonuclease buffer (0.667× PCR buffer, 0.2 μl 10× Exo I buffer, 0.6 μl Exo I (TAKARA), 6.33 mM dithiothreitol). After primer digestion at 37 °C for 30 min and 80 °C for 20 min, the sample was mixed with 5 μl Poly-A tailing buffer (1× PCR buffer, 3 mM dATP, 0.12 μl RNaseH (TAKARA), 0.42 μl TdT enzyme (TAKARA)), subjected to Poly-A tailing reaction at 37 °C for 50 s and 65 °C for 10 min, then mixed with 46 μl Second-strand buffer (1.09× MightyAmp Buffer Ver.2 (TAKARA), 69.6 nM Tagging primer, MightyAmp DNA Polymerase (TAKARA)) and subjected to second-strand synthesis at 98 °C for 130 s, 40 °C for 1 min and 68 °C for 5 min. Finally, the sample was mixed with 50 μl PCR buffer (1× MightyAmp Buffer Ver.2, 1.9 μM PCR primer) and subjected to PCR amplification: 15 cycles of 98 °C for 10 s, 65 °C for 15 s and 68 °C for 5 min with a final extension at 68 °C for 5 min. The amplified cDNA was prepared for sequencing with the use of a Nextera XT DNA Sample Prep Kit (Illumina) and subjected to deep sequencing on the Illumina HiSeq2500 platform to yield 36-base single-end reads. Approximately 2 million sequences were obtained from each sample. Sequences were mapped to the reference mouse genome (mm9) with ELAND v2 (Illumina). Only uniquely mapped tags with no base mismatches were used for the analysis. Gene expression was quantitated as reads per kilobase of exon per million mapped reads (RPKM) on the basis of RefSeq gene models (mm9). *p* value and FDR of each gene whose expression level was >1 RPKM in at least one sample were determined with a paired-comparison experiment of *edgeR* (v3.30.3) of the *R* package. Genes whose Ring1B KO/control and control/Ring1B KO fold changes were ≥1.5 on average and ≥1.2 in each experiment, with an FDR of <0.15 were defined as upregulated and downregulated genes, respectively. KEGG pathways enriched in upregulated and downregulated genes were determined with the use of DAVID Bioinformatic Resources 6.8[52,53].

**RT-qPCR analysis**. Total RNA was isolated from cells with the use of RNAiso plus (Takara), and up to 0.5 μg of the RNA was subjected to RT with the use of ReverTra Ace qPCR RT Master Mix with gDNA remover (Toyobo). The resulting cDNA was subjected to real-time PCR analysis in a LightCycler 480 instrument (Roche) with either KAPA SYBR FAST for LightCycler 480 (Kapa Biosystems) or Thunderbird SYBR qPCR mix (Toyobo). The amount of each target mRNA was normalized by that of *Actb* mRNA. Primer sequences are provided in Supplementary Table 1.

**In situ hybridization analysis**. For the detection of *Axin2*, *Wnt8b* and *Bmp4* mRNA, we used an RNAscope 2.5 HD Reagent Kit-BROWN (Advanced Cell

Diagnostics) according to the manufacturer's protocol[54], but with minor modifications. In brief, coronal sections of embryos (fixed overnight with 4% PFA in PBS) were baked at 60 °C for 30 min, fixed with 4% PFA in PBS on ice, washed with 100% ethanol at room temperature, allowed to dry in air, and treated with 30.0 to 35.5% (w/w) hydrogen peroxide for 10 min at room temperature. The sections were then washed with water, boiled for 5 min in a target retrieval solution (Advanced Cell Diagnostics), washed first with water and then with 100% ethanol, allowed to dry overnight, treated with the protease for 10 min at 40 °C, and exposed to the *Axin2*, *Wnt8b* and *Bmp4* probe (Advanced Cell Diagnostics) for 2 h at 40 °C before signal amplification and detection with 3,3′-diaminobenzidine. Sections were finally washed consecutively with water, 70 and 100% ethanol, and 100% xylene before mounting in EcoMount (Biocare Medical).

For the detection of *Shh* mRNA, a standard protocol was performed. For the preparation of the digoxigenin-labeled riboprobe, the linearized plasmid containing the probe sequence was incubated for 3 h at 37 °C with DIG RNA Labeling Mix, Transcription Buffer, and RNA polymerase (Roche) as well as RNase inhibitor (Toyobo). The plasmid was then digested with DNaseI (Takara) for 30 min at 37 °C, after which the DNase reaction was stopped with Stop Solution (Promega). The riboprobe was purified with the use of a ProbeQuant G-50 column (GE Healthcare) and diluted with hybridization buffer (5× Denhardt's solution, 5× standard saline citrate, 50% formamide, tRNA at 250 μg/ml, salmon testis DNA at 200 μg/ml, heparin at 100 μg/ml, and 0.1% Tween 20). The riboprobe (0.5 μg/ml) was denatured at 85 °C for 5 min, placed on ice for 2 min, and then maintained at 65 °C before in situ hybridization. Coronal sections of embryos (fixed for 3 h with 4% PFA in PBS) were fixed again for 10 min with 4% PFA in PBS, washed with 0.1% Tween 20 in PBS, and incubated at room temperature first with 0.1 M triethanolamine for 3 min and then with the same solution containing 0.1% acetic anhydride for 10 min. They were washed again with 0.1% Tween 20 in PBS before incubation at 65 °C first for 1 h with hybridization buffer and then overnight with the denatured RNA probe in 50% formamide within a humidified box. The sections were washed twice for 30 min at 65 °C with 2× standard saline citrate, twice for 30 min at 65 °C with the same solution containing 50% formamide, and three times for 5 min at room temperature with 0.1% Tween 20 in MAB buffer (MABT). After exposure for 1 h at room temperature to 10% fetal bovine serum in MABT, the sections were incubated overnight at 4 °C with alkaline phosphatase-conjugated antibodies to digoxigenin (Roche) at a dilution of 1:2000 in the same solution, washed twice for 10 min at room temperature with MABT and twice for 10 min at room temperature with a solution containing 100 mM Tris-HCl (pH 9.5), 100 mM NaCl, 50 mM MgCl₂, and 0.02% Tween 20, and then incubated at room temperature in the same solution containing NBT-BCIP (nitrotetrazolium blue chloride at 350 μg/ml and 5-bromo-4-chloro-3-indolyl phosphate *p*-toluidine salt at 175 μg/ml) (Roche) until the color appeared. They were finally washed with 0.1% Tween 20 in PBS and mounted in Mowiol. Images for both *Axin2*, *Wnt8b*, *Bmp4* and *Shh* transcripts were acquired with an Axiovert 200 M microscope fitted with an Axiocam or Axiocam 305 camera (Carl Zeiss) and were processed with ImageJ (NIH).

**ChIP-qPCR analysis**. ChIP for Ring1B and H3K27me3 was carried out as described[40] with minor modifications. Cells were fixed with 1% formaldehyde and stored at −80 °C until analysis. Cells were thawed, suspended in RIPA buffer for sonication (10 mM Tris HCl at pH 8.0, 1 mM EDTA, 140 mM NaCl, 1% Triton X 100, 0.1% SDS, and 0.1% sodium deoxycholate) and subjected to ultrasonic treatment using Picoruptor (15 cycles of 30 s ON and 30 s OFF) (Diagenode). The cell lysates were then diluted with RIPA buffer for immunoprecipitation (50 mM Tris-HCl at pH 8.0, 150 mM NaCl, 2 mM EDTA, 1% Nonidet P 40, 0.1% SDS, and 0.5% sodium deoxycholate) and incubated for 1 h at 4 °C with Protein A/G Magnetic Beads (Pierce) to clear non-specific reactivity. They were then incubated overnight at 4 °C with Protein A/G Magnetic Beads that had previously been incubated for overnight at 4 °C with antibodies. The beads were isolated and washed first three times with wash buffer (2 mM EDTA, 150 mM NaCl, 0.1% SDS, 1% Triton X 100, and 20 mM Tris HCl at pH 8.0) and then once with wash buffer containing 500 mM NaCl. Immune complexes were eluted from the beads for 15 min at 65 °C with a solution containing 10 mM Tris HCl at pH 8.0, 5 mM EDTA, 300 mM NaCl, and 0.5% SDS, and they were then subjected to digestion with proteinase K (Nacalai) for >6 h at 37 °C, removal of cross links by incubation at 65 °C for >6 h, and extraction of the remaining DNA with phenol–chloroform–isoamyl alcohol and ethanol. The DNA was washed with 70% ethanol, suspended in water, and subjected to real-time PCR analysis in a LightCycler 480 (Roche) with Thunderbird SYBR qPCR Mix (Toyobo). Antibodies and primer sequences are presented in Supplementary Tables 6 and 2, respectively.

**CUT&Tag analysis**. CUT&Tag was performed as described[55], with minor modifications. Sorted cells were counted, suspended in CELLBANKER 1 (ZENOAQ), and stored at −80 °C until analysis. Concanavalin A beads (Bangs Laboratories) were washed with binding buffer (10 mM KCl, 1 mM CaCl₂, 1 mM MgCl₂, and 20 mM HEPES at pH 7.5), mixed with thawed cells with HEK293T cells for spike-in control, and rotated for 10 to 20 min at room temperature. After washed with Wash buffer (150 mM NaCl, 0.5 mM spermidine, a protease inhibitor (Roche) and 20 mM HEPES at pH 7.5) once, the beads were suspended in Antibody buffer (150 mM NaCl, 0.5 mM spermidine, a protease inhibitor, 0.05% digitonin, 2 mM EDTA,

0.1% BSA, and 20 mM HEPES at pH 7.5) with primary antibodies for overnight at 4 °C. After washed with Dig-wash buffer (150 mM NaCl, 0.5 mM spermidine, a protease inhibitor, 0.05% digitonin, and 20 mM HEPES at pH 7.5) once, the beads were suspended in Dig-wash buffer with secondary antibodies for 30 min at room temperature. The beads were washed with Dig-wash buffer for three times and subjected to binding reaction of pAG-Tn5 adaptor complex in Dig-300 buffer (300 mM NaCl, 0.5 mM spermidine, protease inhibitor, 0.01% digitonin, and 20 mM HEPES at pH 7.5) for 1 h at room temperature. Note that pAG-Tn5 was constructed with the use of both pAG-MNase (Addgene #123461) and pA-Tn5 (Addgene #124601) plasmids and was used instead of pA-Tn5 in the original protocol. After washed with Dig-300 buffer for four times, the beads were suspended with Tagmentation buffer (300 mM NaCl, 0.5 mM spermidine, a protease inhibitor, 0.01% digitonin, 1 mM MgCl$_2$, and 20 mM HEPES at pH 7.5) for tagmentation reaction for 1 h at 37 °C. The tagmented DNA was eluted by adding 0.5 M EDTA (final 17 mM), 10% SDS (final 0.1%), and 10 mg/ml proteinase K (final 0.17 mg/ml) for overnight at 37 °C, purified using phenol–chloroform–isoamyl alcohol and ethanol, washed with 70% ethanol and suspended in water. The DNA was amplified by PCR reaction using Q5 Hot Start High-Fidelity 2x Master Mix (NEB) as follows: 5 min at 72 °C and 30 s at 98 °C followed by 18 cycles of 10 s at 98 °C and 20 s at 63 °C, with a final extension at 72 °C for 1 min. Finally, the amplified DNA was purified using SPRI beads (Beckman Coulter). Antibodies are listed in Supplementary Table 7.

For normalization of the H3K27me3 CUT&Tag data with the use of scaling factors, the precise fraction of the spike-in human genome relative to the mouse genome was determined. A sample corresponding to 10% of the mixture of mouse and human cells used for CUT&Tag was thus subjected to DNA extraction, and DNA libraries were prepared from the extracted DNA with the use of a Nextera XT DNA Library Prep Kit (Illumina). The libraries for CUT&Tag and DNA-seq were sequenced as 50- and 150-bp paired-end reads on the Illumina NovaSeq 6000 and HiSeq X platforms, respectively. Sequencing results were mapped to the hybrid genome of mouse (mm10) and human (hg38) with the use of Bowtie2[69], and coverage scores for fragments that mapped uniquely to mm10 were quantified with Deeptools 3.3.1[70].

**Primary culture of the telencephalon.** Monolayer culture was performed as described[40]. Primary NPCs were isolated from the CTX of ICR mouse embryos at E9. Dissected cortices were subjected to enzymatic digestion with a papain-based solution (FUJIFILM Wako Chemicals), and the dissociated cells were cultured in Dulbecco's modified Eagle's medium (DMEM)–F12 (1:1, v/v) supplemented with B27 (Invitrogen) and recombinant human FGF2 (20 ng/ml) (Invitrogen) on poly-D-lysine coated dishes. For explant culture, the dissected telencephalon of ICR mouse embryos was cultured in DMEM–F12 supplemented with B27 and recombinant human FGF2 (20 ng/ml). After culture of cells or explant tissue for 6 h, half of the medium was removed and replaced with medium supplemented with B27, human FGF2, and either SAG (Sigma-Aldrich), recombinant human BMP4 (R&D Systems), CHIR-99201 (Wako), or cyclopamine (Enzo Life Sciences). SAG was dissolved in dimethyl sulfoxide at a concentration of 5 mM and was added to the culture medium at a final concentration of 2 µM. Recombinant BMP4 was dissolved at a concentration of 100 µg/ml in sterile 4 mM HCl containing 0.1% BSA and was added to the culture medium at a final concentration of 50 ng/ml. CHIR-99201 was dissolved in dimethyl sulfoxide at a concentration of 1 mM and was added to the culture medium at a final concentration of 5 µM. Cyclopamine was dissolved in ethanol at a concentration of 1 mM and was added to the culture medium at a final concentration of 1 µM. The cells or tissue were cultured for an additional 24 h and then frozen in liquid nitrogen before analysis.

**Statistical analysis.** Data are presented as means ± s.d. and were compared with the two-tailed Student's paired or unpaired $t$ test or by analysis of variance (ANOVA) followed by either the Benjamini–Hochberg multiple-comparison test or Dunnett's multiple-comparison test. Statistical assessment of RNA-seq and KEGG pathway analyses was performed with the use of *edgeR* and DAVID Bioinformatic Resources[52,53].

**Reporting summary.** Further information on research design is available in the Nature Research Reporting Summary linked to this article.

## Data availability
The data that support this study are available from the corresponding authors upon reasonable request. The sequence data have been deposited in the DNA Data Bank of Japan (DDBJ) Sequence Read Archive under the following accession codes: DRA008366 (Quartz-seq analysis for NPCs of Ring1B KO embryos), DRA010033 (CUT&Tag analysis of H3K27me3 for NPCs from DM, CTX, and V regions of the mouse telencephalon), and DRA010296 (CUT&Tag analysis of Ring1B for NPCs from DM, CTX, and V regions of the mouse telencephalon). WIG files for CUT&Tag analysis have also been deposited in the DDBJ Genomic Expression Archive under the accession code: E-GEAD-378 [ftp://ftp.ddbj.nig.ac.jp/ddbj_database/gea/experiment/E-GEAD-000/]. Source data are provided with this paper.

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

## Acknowledgements

We thank S. Nishikawa (RIKEN) for providing Sox1-Cre mice; I. Nikaido (RIKEN) for tips on performance of the Quartz method; K. Imamura, T. Horiuchi, S. Sugano, and Y. Suzuki (The University of Tokyo) for performing high-throughput sequencing analysis; M. Endoh (NUS) for tips on performance of the Ring1B ChIP assay; Y. Koseki (RIKEN) for providing Ring1 mutant mice; the One-Stop Sharing Facility Center for Future Drug Discoveries (The University of Tokyo) for FACS; Y. Maeda, R. Nagayoshi, and Y. Kakeya for technical assistance; and members of the Gotoh laboratory for discussion. This research was supported by AMED-CREST (JP20gm1310004 to Y.G.), AMED-PRIME (JP20gm6110021 to Y.K.), and MEXT/JSPS KAKENHI (JP20J14522 to H.E.; JP16H06481 and JP16H06479 to Y.G.; JP19H05253, JP20H03179, JP20H05383, and JP16H06279 to Y.K.) grants.

## Author contributions

H.E., Y.K. and Y.G. designed the study and wrote the manuscript. H.E., Y.K., N.Y.-K., H.S., and S.U. performed the experiments and analyzed the data. H.K. generated $Ring1a^{-/-}$; $Ring1b^{flox/flox}$ mice. Y.K. and Y.G. supervised the study. All authors contributed to revision and approved the final version of the manuscript.

## Competing interests

The authors declare no competing interests.
