## [Peer Review File · Nature Communications]

Reviewers' Comments:

Reviewer #1:

Remarks to the Author:

In this manuscript, Eto et al. study the role of the Polycomb group proteins Ring1A/B in the early developing mouse telencephalon using a conditional knockout approach. Deletion of Ring1A/B resulted in dorsalized expression of various transcription factors in neural progenitor cells of the ventral telencephalon, indicating that Polycomb group proteins are not only involved in the maintenance of the A-P axis during development, but also contribute to the regulation of dorso-ventral patterning during the development of the early telencephalon. Transcriptome analysis revealed that Ring1B suppresses several major signaling pathways, including Wnt and BMP pathways, resulting in reduced Shh expression. Eto et al. went on to show that H3K27me3 and Ring1b are found at the promoters of a subset of BMP and Wnt pathway genes, providing a mechanism for the action of Ring1 proteins in early neural progenitor cells.

Overall, the study is well executed and the manuscript well written. The results contribute to a better understanding of the regulation of early telencephalon development and in particular, the role of Polycomb group-mediated regulation in this process. I find the study interesting and suitable for publication in Nature Communications.

Nevertheless, I have some points that should be taken into consideration:

- 1) The authors mention that the Sox1-Cre is expressed from <E8.5 and in combination with Ring1b F/F induces dorsalization of the early telencephalon (p. 6). In contrast, the Foxg1-IRES-Cre, expressed from E9.0, does not lead to this phenotype (data not shown, discussed on p. 13). I think would be helpful to include the data. Moreover, it would be important to know when the Ring1 proteins are lost. Currently, data is presented for E10, but since half a day appears to make a difference, it would be interesting to know when Ring1 action is required. Indeed, already at the very early neuroepithelial stage?
- 2) In addition, in Figure S1A it appears as if there was a gradient with Ring1B staining showing a greater reduction in the ventral region of the telencephalon. Is this a representative image?
- 3) The authors state (on p. 13) that they did not detect premature neurogenesis in the KO embryos (data not shown). It would be important to show the data as this is an alternative potential mechanism leading to a reduced brain size.
- 4) In addition to Ring1b, H3K27me3 is strongly enriched at several Wnt and BMP pathway genes (Figure 8B). Given that the classical model of consecutive PcG action (PRC2 => H3K27me3 => PRC1) has been repeatedly challenged, it would be interesting to know whether H3K27me3 levels are affected at Wnt and BMP genes upon Ring1 KO.
- 5) While the Control and Ring1B KO experiments are appropriately quantified, the IHC for Ring1A and Ring1A/B KO lacks quantification (Figures 1, 2, 3, 6). The quantification should be added.
- 6) Regarding the statistical analysis of experimental vs. control conditions, throughout the paper, the authors present data as Mean +/- SEM. Would the SD (standard deviation) not be more appropriate? Moreover, for several experiments (Figure 1, 2, 3, 4, 7), a paired t-test is applied, which I consider not justified as the animals are not subject to repeated measuring.
- 7) Regarding the gene expression analysis, an FDR of 0.15 was chosen. This appears very high considering that 15% of genes might be false positives. What was the rationale for this cut-off?
- 8) The authors present a list of genes differentially expressed upon Ring1b KO, however, the full RNA-seq data is not provided. Deposition of the RNA-seq data in the GEO repository and addition

of the accession number would increase the value of the data set for the research community.

9) In Figure 4F, the 'Relative mRNA amount' of Wnt and Bmp pathway genes was determined by RT-qPCR. The KO conditions lacks error bars. Or was this KO set to a value of 1. This would be rather unconventional.

10) In Figure 5A and B, the in situ hybridization signal is very difficult to see. Could the presentation be improved, possibly by showing even higher magnification zooms?

11) There is a typo in line 385 ('independent').

Reviewer #2:

Remarks to the Author:

In this manuscript, Eto et al have investigated the role of the Polycomb group (PcG) protein Ring1 in mouse cortical development. The authors deleted Ring1b in the early neuroepithelium with the Sox1-Cre transgene, and by embryonic day 11 (E11), the telencephalon was significantly smaller. Apoptosis was significantly increased in Ring1b knockout (KO) mice, as well as the Ring1a/b double KO (dKO) mice. Because the brain was already significantly smaller by E11, the authors then focused most studies to E10-11. At E10, expression of ventral genes Nkx2.1 and Gsx2 was decreased in Ring1 KO mice. Ring1b KO mice also had increased expression of dorsal genes such as Emx1 in ventral brain regions. RNA-seq analysis of E11 Ring1b KO ventral brain revealed 953 upregulated genes, including those related to BMP and Wnt signaling. RNA-seq analysis also revealed 238 down-regulated genes including those related to Shh signaling. Addition of Shh agonist (SAG) to E10 Ring1b KO cultures caused upregulation of Shh-responsive genes such as Gli1 and Ptch1. Addition of BMP and Wnt activators to these cultures suppressed Shh expression. Blocking Shh signaling with cyclopamine did not affect BMP or Wnt ligands. ChIP-qPCR assays showed increased levels of H3K27me3 (a mark of polycomb activity) at Bmp4, Bmp7, Wnt 7b and Wnt8b. ChIP-qPCR showed Ring1B protein at Bmp4 and Wnt8b.

The authors clearly show that the genetic loss of Ring1b causes significant developmental phenotypes in the telencephalon (in this report, the brain is already much smaller by E11 and is accompanied by increased cell death). However, it is not very surprising that deletion of a Polycomb factor very early in the neuroepithelium would result in profound effects. From our understanding of Ring1 as part of the polycomb-repressive complex, it is already clear that Ring1b deletion would lead to de-repression of many important genes. In this paper, Ring1b KO caused depression of BMP and Wnt ligands, which likely contributed to the mis-expression of dorsal genes in ventral brain regions (and possibly also the loss of ventral gene expression). The authors have focused on genes related to dorsoventral patterning, but the data do not show that Ring1b "regulates" such patterning. Rather, the data simply show that Ring1 proteins are required for proper gene expression, including that related to dorsoventral patterning. Overall, these results seem fairly incremental to our knowledge of Ring1 function in neural development as well as its role as a transcriptional regulator.

Other comments:

1. Page 9: The subheading, "Ring1 promotes Shh expression and activates the Shh signaling pathway..." is an over-statement of the results. While Ring1b deletion causes loss of Shh expression, this result does not indicate that Ring1b "promotes" Shh expression. Indeed, the loss of Shh expression is likely quite indirect, as the authors show that Ring1b KO causes de-repression of BMP and Wnt ligands, and pharmacological activation of BMP/Wnt signaling reduced Shh expression in cultured cells.
2. In the abstract, the authors write, "Our results indicate that PcG suppresses BMP and Wnt in a region-specific manner so that Shh can be expressed properly..." The ChIP data do not strongly

support this conclusion. The authors performed ChIP-qPCR analysis of E9 telencephalic tissues (not region-specific), and found that Ring1b and H3K27me3 are found at BMP/Wnt genes. This suggests that BMP/Wnt genes are targeted by PcG proteins throughout the brain. Do PcG proteins only target BMP and Wnt genes in ventral brain? If so, how is this regional-specificity achieved?

3. Does Ring1b deletion lead to loss of H3K27me3 and Ring1b at BMP and Wnt genes? This would be a much better control for these experiments. The authors should also consider performing H3K27me3 (and H2AK119ub) ChIP-seq, to analyze the chromatin changes that result from the early loss of Ring1b. This would also help distinguish which gene expression changes are more directly related to loss of Ring1b-mediated repression, from those that are indirectly related.

Reviewers' comments:

Reviewer #1 (Remarks to the Author):

In this manuscript, Eto et al. study the role of the Polycomb group proteins Ring1A/B in the early developing mouse telencephalon using a conditional knockout approach. Deletion of Ring1A/B resulted in dorsalized expression of various transcription factors in neural progenitor cells of the ventral telencephalon, indicating that Polycomb group proteins are not only involved in the maintenance of the A-P axis during development, but also contribute to the regulation of dorso-ventral patterning during the development of the early telencephalon. Transcriptome analysis revealed that Ring1B suppresses several major signaling pathways, including Wnt and BMP pathways, resulting in reduced Shh expression. Eto et al. went on to show that H3K27me3 and Ring1b are found at the promoters of a subset of BMP and Wnt pathway genes, providing a mechanism for the action of Ring1 proteins in early neural progenitor cells.

Overall, the study is well executed and the manuscript well written. The results contribute to a better understanding of the regulation of early telencephalon development and in particular, the role of Polycomb group-mediated regulation in this process. I find the study interesting and suitable for publication in Nature Communications.

We thank the reviewer for his/her strong support on our study and constructive criticisms, which we address in detail below.

Nevertheless, I have some points that should be taken into consideration:

1) The authors mention that the Sox1-Cre is expressed from <E8.5 and in combination with Ring1b F/F induces dorsalization of the early telencephalon (p. 6). In contrast, the Foxg1-IRES-Cre, expressed from E9.0, does not lead to this phenotype (data not shown, discussed on p. 13). I think would be helpful to include the data. Moreover, it would be important to know when the Ring1 proteins are lost. Currently, data is presented for E10, but since half a day appears to make a difference, it would be interesting to know when Ring1 action is required. Indeed, already at the very early neuroepithelial stage?

As suggested by the reviewer, we now include the data of Ring1B deletion with the use of the *Foxg1-IRES-Cre* mice (new Supplementary Fig. 10). We found that the proportion of the dorsal (*Pax6*⁺ and *Neurog1*⁺) and ventral (*Nkx2.1*⁺ and *Ascl1*⁺) telencephalic regions was not significantly different between these Ring1B-depleted mice and control mice at E11 (new

Supplementary Fig. 10), indicating that the dorsalization phenotype was not found when the *Ring1B* gene was deleted by *Foxg1-IRES-Cre*. These new data are now mentioned in our revised discussion (page 13, lines 415-420).

Moreover, as suggested, we examined the time of reduction of Ring1B protein when the *Ring1B* gene was deleted by *Sox1-Cre* (new Supplementary Fig. 1a–d, h, i, l, m). We found that Ring1B proteins and H2AK119ub were already reduced in the telencephalon at E9 when the *Ring1B* gene was deleted by *Sox1-Cre* in control mice as well as in Ring1A KO mice (new Supplementary Fig. 1h–o). These results suggest a role of Ring1B action at the early developmental stage in establishing the dorsoventral patterning of telencephalon. These new data are now mentioned in our revised results (page 5, lines 131-144).

2) In addition, in Figure S1A it appears as if there was a gradient with Ring1B staining showing a greater reduction in the ventral region of the telencephalon. Is this a representative image?

No, this apparent gradient is not representative. We thank the reviewer for pointing this out. We thus replaced the images with more representative ones (new Supplementary Fig. 1a, b).

Moreover, we quantified the level of Ring1B immunohistofluorescence signals along the dorsoventral axis of telencephalon (new Supplementary Fig. 1c, d). The Ring1B signal intensities appeared quite even along this axis in control telencephalon as well as in Ring1A KO telencephalon. These new data are now mentioned in our revised results and discussion (page 5, lines 136-138; page 13, lines 433-435).

3) The authors state (on p. 13) that they did not detect premature neurogenesis in the KO embryos (data not shown). It would be important to show the data as this is an alternative potential mechanism leading to a reduced brain size.

We agree with the reviewer that this is an important point. We now include the results regarding the effect of Ring1B deletion on the extent of neurogenesis at E9, when telencephalic NPCs start neurogenesis in BL6/J mice (new Supplementary Fig. 9). At this stage, a small number of cells were found to become positive for the neuronal marker β III-tubulin in Ring1A-deleted telencephalon, in particular in the ventrocaudal part. Importantly, we found that Ring1B deletion did not increase β III-tubulin⁺ cells in Ring1A-deleted telencephalon, indicating that it does not cause premature neurogenesis in the early-stage telencephalon. These new data are now described in our revised discussion (page 13, lines 411-414).

4) In addition to Ring1b, H3K27me3 is strongly enriched at several Wnt and BMP pathway genes (Figure 8B). Given that the classical model of consecutive PcG action (PRC2 => H3K27me3 =>

PRC1) has been repeatedly challenged, it would be interesting to know whether H3K27me3 levels are affected at Wnt and BMP genes upon Ring1 KO.

We thank the reviewer for raising this excellent point with regard to the effect of Ring1/PRC1 on H3K27me3. We thus investigated the levels of H3K27me3 at the *Bmp4*, *Bmp7*, *Wnt7b* and *Wnt8b* loci as well as those at the *β-actin* or *Gapdh* loci (negative controls) and the *Hoxa1* or *Hoxd3* loci (positive controls) by ChIP-qPCR in control and Ring1B-deleted telencephalon at E10 and, unexpectedly, found that Ring1B deletion did not significantly reduce the levels of H3K27me3 at these loci except for the *Bmp7* locus (new Supplementary Fig. 7). This is a very important observation given that Ring1B deletion under the same condition was sufficient for the increase of their gene expression (new Fig. 4f, g, 5a–d, h–k). These results suggest that H3K27me3 deposition alone is not sufficient for suppression of these BMP and Wnt gene loci while Ring1B is required for their suppression. This is consistent with a previously-proposed notion that PRC1 rather than PRC2 is responsible for gene suppression (Fursova et al., 2019; Blackledge et al., 2020). These new data are now described in our revised results (page 11, lines 340-344) and highlighted in discussion (page 14, lines 446-457).

5) While the Control and Ring1B KO experiments are appropriately quantified, the IHC for Ring1A and Ring1A/B KO lacks quantification (Figures 1, 2, 3, 6). The quantification should be added.

According to the suggestion, we added quantification of the immunohistofluorescence signals for Nkx2.1, Pax6 and Neurog1 proteins (new Fig. 2e, f, k, 3g). We did not quantify the signals for Ascl1 protein and *Shh* mRNA in new Fig. 3f and 6e because they were under detectable levels in Ring1A/B dKO telencephalon.

6) Regarding the statistical analysis of experimental vs. control conditions, throughout the paper, the authors present data as Mean +/- SEM. Would the SD (standard deviation) not be more appropriate? Moreover, for several experiments (Figure 1, 2, 3, 4, 7), a paired t-test is applied, which I consider not justified as the animals are not subject to repeated measuring.

According to the suggestion, we changed SEM to SD throughout the paper.

As regards the statistical analyses, we used paired *t* test in several experiments because of the following reasons. In new Fig. 1f and 1h, the size of telencephalon was compared between control and Ring1B-deleted embryos among littermates, and we analyzed four litters. Given that the conditions between littermates are more similar than those between nonlittermates (with respect to environmental conditions and developmental age), we employed paired *t* test. In the experiments of new Fig. 2, 3, 4 and 7, the data were compared between control and Ring1B-

deleted embryos among littermates, and we analyzed several litters. Moreover, the experiments such as immunostaining, chromatin immunoprecipitation and RT-qPCR analyses were performed separately for each litter. Again, given that the conditions between littermates are expected to be more similar than those between nonlittermates, we employed paired *t* test.

7) Regarding the gene expression analysis, an FDR of 0.15 was chosen. This appears very high considering that 15% of genes might be false positives. What was the rationale for this cut-off?

Even though we chose the cut-off of 0.15 for FDR, *p*-values of individual genes were sufficiently low, with the highest *p*-value being 0.0173 among the genes with a FDR of <0.15 in new Supplementary Table 1. This is based on the definition of FDR by which the information of all genes is taken into account to evaluate a trend of many genes simultaneously (which is good for some analyses such as the pathway analysis we carried out in Fig. 4). Indeed, a number of papers chose an FDR of <0.15 (or a higher cut-off level) (e.g. a list of references).

Moreover, importantly, we confirmed the changes in expression of genes (related to Shh, BMP and Wnt signaling pathways by Ring1B deletion) picked up by this cut-off of FDR in the RNA-seq analysis with other means such as RT-qPCR (new Fig. 4f, g, 6b), in situ hybridization (new Fig. 5a–k) and immunohistochemical (Fig. 5l–n) analyses.

Reference list

1. Shamsi, F. *et al.* FGF6 and FGF9 regulate UCP1 expression independent of brown adipogenesis. *Nature Communications* **11**, 1421 (2020).
2. Liu, Z. *et al.* CASZ1 induces skeletal muscle and rhabdomyosarcoma differentiation through a feed-forward loop with MYOD and MYOG. *Nature Communications* **11**, 911 (2020).
3. Basilico, S. *et al.* Dissecting the early steps of MLL induced leukaemogenic transformation using a mouse model of AML. *Nature Communications* **11**, 1407 (2020)
4. Tang, S. J. *et al.* Cis- And Trans-Regulations of pre-mRNA Splicing by RNA Editing Enzymes Influence Cancer Development. *Nature Communications* **11**, 799 (2020)

8) The authors present a list of genes differentially expressed upon Ring1b KO, however, the full RNA-seq data is not provided. Deposition of the RNA-seq data in the GEO repository and addition of the accession number would increase the value of the data set for the research community.

Indeed, we agree that making datasets public is extremely important. We have uploaded the datasets of RNA-seq and CUT&Tag analyses to the DDBJ server. Accession numbers are DRA008366 (Quartz-seq analysis on NPCs of Ring1B KO embryos), DRA010033 (CUT&Tag

analysis of H3K27me3 in NPCs derived from DM, CTX, and V regions of the mouse telencephalon), and DRA010296 (CUT&Tag analysis of Ring1B in NPCs derived from DM, CTX, and V regions of the mouse telencephalon). The data will become open upon publication of this paper.

9) In Figure 4F, the 'Relative mRNA amount' of Wnt and Bmp pathway genes was determined by RT-qPCR. The KO conditions lacks error bars. Or was this KO set to a value of 1. This would be rather unconventional.

We agree with the reviewer and thus set the controls to a value of 1 in new Fig. 4f, g (instead of KO conditions).

10) In Figure 5A and B, the in situ hybridization signal is very difficult to see. Could the presentation be improved, possibly by showing even higher magnification zooms?

We appreciate the reviewer for pointing this out. To improve the quality of the images, we performed RNAscope®, a highly sensitive method for *in situ* hybridization (new Fig. 5a–k). We could detect clearer signals of *Wnt8b*, *Axin2* and *Bmp4* mRNA in mouse telencephalons at E10 and confirmed a ventral expansion of these mRNA by Ring1B deletion in the Ring1A KO background. Images at a higher magnification are also shown (new Fig. 5b, c, f, i, j). These new data are now mentioned in our revised results (page 8–9, lines 252-267).

11) There is a typo in line 385 ('independent').

We thank the reviewer for catching this typo and correct it.

We really appreciate the constructive comments from this reviewer. With the substantial amount of new data incorporated, we feel that our manuscript is significantly improved.

--

Reviewer #2 (Remarks to the Author):

In this manuscript, Eto et al have investigated the role of the Polycomb group (PcG) protein Ring1 in mouse cortical development. The authors deleted Ring1b in the early neuroepithelium with the Sox1-Cre transgene, and by embryonic day 11 (E11), the telencephalon was significantly smaller. Apoptosis was significantly increased in Ring1b knockout (KO) mice, as well as the Ring1a/b double KO (dKO) mice. Because the brain was already significantly smaller by E11, the authors then focused most studies to E10-11. At E10, expression of ventral genes Nkx2.1 and

Gsx2 was decreased in Ring1 KO mice. Ring1b KO mice also had increased expression of dorsal genes such as Emx1 in ventral brain regions. RNA-seq analysis of E11 Ring1b KO ventral brain revealed 953 upregulated genes, including those related to BMP and Wnt signaling. RNA-seq analysis also revealed 238 down-regulated genes including those related to Shh signaling. Addition of Shh agonist (SAG) to E10 Ring1b KO cultures caused upregulation of Shh-responsive genes such as Gli1 and Ptch1. Addition of BMP and Wnt activators to these cultures suppressed Shh expression. Blocking Shh signaling with cyclopamine did not affect BMP or Wnt ligands. ChIP-qPCR assays showed increased levels of H3K27me3 (a mark of polycomb activity) at Bmp4, Bmp7, Wnt 7b and Wnt8b. ChIP-qPCR showed Ring1B protein at Bmp4 and Wnt8b.

The authors clearly show that the genetic loss of Ring1b causes significant developmental phenotypes in the telencephalon (in this report, the brain is already much smaller by E11 and is accompanied by increased cell death). However, it is not very surprising that deletion of a Polycomb factor very early in the neuroepithelium would result in profound effects. From our understanding of Ring1 as part of the polycomb-repressive complex, it is already clear that Ring1b deletion would lead to de-repression of many important genes. In this paper, Ring1b KO caused depression of BMP and Wnt ligands, which likely contributed to the mis-expression of dorsal genes in ventral brain regions (and possibly also the loss of ventral gene expression). The authors have focused on genes related to dorsoventral patterning, but the data do not show that Ring1b “regulates” such patterning. Rather, the data simply show that Ring1 proteins are required for proper gene expression, including that related to dorsoventral patterning. Overall, these results seem fairly incremental to our knowledge of Ring1 function in neural development as well as its role as a transcriptional regulator.

We thank the reviewer for constructive criticisms, which we address in detail below.

Although it has been reported that Ring1/PcG regulates many developmental genes, this study provides the very first example to show that Ring1/PcG regulates morphogen expression in an area-specific manner.

Morphogenetic signals and their downstream transcription factors determine regional identity along the D-V axis in the developing central nervous system. Mutual inhibition between dorsal and ventral transcription factors plays a pivotal role in segregation and maintenance of regional identity, but the mechanisms that underlie the initial regional confinement of morphogen expression have remained largely unknown. We have now found that Ring1/PRC1 plays an essential role in establishment of the spatial expression patterns of morphogens along the D-V axis and in consequent regionalization of the telencephalon at the early stage of mouse development. Our new results (new Fig. 9 and Supplementary Fig. 8) also indicate that Ring1B and H3K27me3 are associated with genes encoding dorsal morphogens—such as *Bmp4*, *Bmp7*,

Wnt7b, and *Wnt8b*—selectively in NPCs localized outside of the dorsal midline, with this specificity possibly accounting for restriction of the expression of these gene to the dorsal region. Our results thus unveil an epigenetic foundation of initial D-V patterning of morphogens in the mouse telencephalon.

In this revised paper, we also newly found that Ring1B deletion results in derepression of its target genes (e.g. *Bmp4* and *Wnt8b*) without reducing the levels of H3K27me3 (new Fig. 4f, g, new Fig. 5a–d,h–k and new Supplementary Fig. 7). This indicates that H3K27me3 alone is not sufficient for the repression of PcG target genes in this developmental context. Although a similar notion has been proposed in the context of mouse ES cells and their differentiation (Riising et al. 2014 showed that PRC2 depletion does not derepress PcG target genes in ES cells; Blackledge et al. 2020 showed that inactivation of Ring1B derepress PcG target genes when the reduction of H3K27me3 was partial), our finding in this paper is probably the first case to show the insufficiency of PRC2 for gene repression in an *in vivo* (developmental) context.

Other comments:

1. Page 9: The subheading, “Ring1 promotes Shh expression and activates the Shh signaling pathway...” is an over-statement of the results. While Ring1b deletion causes loss of Shh expression, this result does not indicate that Ring1b “promotes” Shh expression. Indeed, the loss of Shh expression is likely quite indirect, as the authors show that Ring1b KO causes de-repression of BMP and Wnt ligands, and pharmacological activation of BMP/Wnt signaling reduced Shh expression in cultured cells.

We agree with the reviewer and changed the subheading into “Loss of Ring1 results in downregulation of Shh and its signaling pathway” just to describe the observation (page 9, lines 269-270).

2. In the abstract, the authors write, “Our results indicate that PcG suppresses BMP and Wnt in a region-specific manner so that Shh can be expressed properly...” The ChIP data do not strongly support this conclusion. The authors performed ChIP-qPCR analysis of E9 telencephalic tissues (not region-specific), and found that Ring1b and H3K27me3 are found at BMP/Wnt genes. This suggests that BMP/Wnt genes are targeted by PcG proteins throughout the brain. Do PcG proteins only target BMP and Wnt genes in ventral brain? If so, how is this regional-specificity achieved?

We thank the reviewer for pointing out this very important issue. We thus examined regional differences of Ring1B and H3K27me3 deposition at the *Bmp4* and *Wnt8b* gene loci by dissecting out the dorsal midline (DM), cerebral cortex (CTX) and ventral (V) regions from E11 mouse

telencephalon and isolating NPCs as CD133⁺ cells by FACS from each region (new Fig. 9 and Supplementary Fig. 8). We actually obtained only $\sim 2 \times 10^4$ CD133⁺ cells from the DM telencephalic region from one embryo at E11. It is challenging to reliably detect the deposition patterns of Ring1B protein and H3K27me3 with the use of this small number of cells. In order to overcome this problem, instead of performing a regular ChIP-seq analysis, we employed a CUT&Tag sequencing analysis, a more sensitive method to detect chromatin association of a protein-of-interest (Kaya-Okur et al., 2019). Importantly, we found that the deposition of Ring1B protein as well as that of H3K27me3 at *Bmp4* and *Wnt8b* gene loci was significantly lower in the DM region compared to the CTX and V regions, whereas the levels of Ring1B and H3K27me3 deposition at *Hoxa1*, *Hoxd3*, *Actb* and *Gapdh* gene loci did not show significant differences between the telencephalic regions (new Fig. 9 and Supplementary Fig. 8). The striking regional differences in Ring1B and H3K27me3 deposition may thus account for the region-specific suppression of these morphogens, which may contribute to the establishment of the dorso-ventral patterning of mouse telencephalon. These findings were further supported by a region-specific derepression of *Bmp4* and *Wnt8b* expression in Ring1-deleted mice (new Fig. 5a–d, h–k). These new data are now described in our revised results (page 8–9, lines 252-267; page 11–12, 345-372) and highlighted in discussion (page 12, lines 379-386).

As regards the mechanism that underlies the region-specific deposition of PcG proteins at specific gene loci, this has been a major question in the field of PcG-mediated regulation. We believe that it is beyond the scope of this paper, albeit we understand its importance.

3. Does Ring1b deletion lead to loss of H3K27me3 and Ring1b at BMP and Wnt genes? This would be a much better control for these experiments. The authors should also consider performing H3K27me3 (and H2AK119ub) ChIP-seq, to analyze the chromatin changes that result from the early loss of Ring1b. This would also help distinguish which gene expression changes are more directly related to loss of Ring1b-mediated repression, from those that are indirectly related.

As suggested by the reviewer, we asked whether Ring1B deletion leads to loss of H3K27me3 at BMP and Wnt gene loci. By performing a ChIP-qPCR analysis for H3K27me3, we found that Ring1B deletion with the use of Sox1-Cre did not significantly reduce the amounts of H3K27me3 bound to the *Bmp4*, *Wnt7b* and *Wnt8b* loci at E10 telencephalon (new Supplementary Fig. 7). Given that we observed derepression (increase in expression) of these genes under the same condition, we concluded that H3K27me3 deposition alone is not sufficient for repression of Ring1B/PcG target genes, whereas Ring1B deposition is necessary for this repression, in this developmental context. These new data are now described in our revised results (page 11, 340-344) and highlighted in discussion (page 14, lines 446-457).

Regarding the levels of Ring1B protein and H2AK119ub, given that we observed their global reduction in Ring1B-deleted telencephalon (new Fig. 1a–d, Supplementary Fig. 1a–d), we did not perform their CHIP analysis.

We would like to thank this reviewer for her/his very constructive suggestions. We think we now have a much improved manuscript with incorporation of new data from these suggested experiments.

Reviewers' Comments:

Reviewer #1:

Remarks to the Author:

The revised version of the manuscript includes several new analysis and experiments, strengthening the manuscript and providing more detailed insight into the region-specific regulation of Ring1 target genes. The results contribute to a better understanding of the role of Polycomb proteins during development, in particular, of the early patterning of the nervous system. Moreover, the transcriptome and chromatin data have now been deposited in a repository facilitating in-depth exploration by readers. The author's response to my comments is satisfactory and I therefore recommend the publication of the manuscript "The Polycomb group protein Ring1 regulates dorsoventral patterning of the mouse telencephalon" in Nature Communications.